# A CCG expansion in *ABCD3* causes oculopharyngodistal myopathy in individuals of European ancestry

Oculopharyngodistal myopathy (OPDM) is an inherited myopathy manifesting with ptosis, dysphagia and distal weakness. Pathologically it is characterised by rimmed vacuoles and intranuclear inclusions on muscle biopsy. In recent years CGG • CCG repeat expansion in four different genes were identified in OPDM individuals in Asian populations. None of these have been found in affected individuals of non-Asian ancestry. In this study we describe the identification of CCG expansions in *ABCD3*, ranging from 118 to 694 repeats, in 35 affected individuals across eight unrelated OPDM families of European ancestry. *ABCD3* transcript appears upregulated in fibroblasts and skeletal muscle from OPDM individuals, suggesting a potential role of over-expression of CCG repeat containing *ABCD3* transcript in progressive skeletal muscle degeneration. The study provides further evidence of the role of non-coding repeat expansions in unsolved neuromuscular diseases and strengthens the association between the CGG • CCG repeat motif and a specific pattern of muscle weakness.

Oculopharyngodistal myopathy (OPDM) was first delineated and described in four families presenting with autosomal dominant disease by Satoyoshi and Kinoshita in 1977[1]. Individuals with OPDM typically present with adult-onset progressive ptosis, external ophthalmoplegia and weakness of the facial, masseter and pharyngeal muscles, resulting in dysphagia and weakness of the distal limb muscles. Muscle biopsies show chronic myopathic changes, including the presence of rimmed vacuoles and myeloid bodies within the myoplasm and intranuclear filamentous inclusions[2]. These intranuclear inclusions are also evident in skin biopsies[3].

It would be more than 40 years before the genetic aetiology of OPDM started to be untangled. To date, heterozygous CGG • CCG repeat expansions in the 5′-UTR of four genes (*LRP12*[4], *GIPC1*[5] *NOTCH2NLC*[4], and *RILPL1*[6,7]) have been associated with OPDM in Asian populations, either isolated or as part of a more complex neurological condition[8–10]. However, as yet no OPDM case of European origin has been found to carry CGG repeat expansions at any of the previously identified OPDM loci.

In this work we sought to identify the genetic cause of the OPDM disease in multiple families of European ancestry. Herein we describe and characterise eight unrelated OPDM families, encompassing 35 affected individuals, and identify CCG expansions in the 5′-UTR of *ABCD3* as the cause of cranial and distal limb weakness in all families. Additionally, we identified an increased expression of the CGG expansion-containing *ABCD3* transcript, as a possible disease mechanism underlying muscle degeneration.

## Results

We report 35 OPDM individuals from eight unrelated families from Australia, the UK and France presenting with OPDM, including six with clear autosomal dominant inheritance (Fig. 1) and present detailed clinical information for 24 affected individuals (Table 1).

### Identification of CGG repeat expansion in the *ABCD3* 5′-UTR in European OPDM

Combined linkage performed for two of the families, AUS1 (*n* = 5 affected individuals) and AUS2 (*n* = 4 affected individuals), produced a maximum combined multipoint logarithm of odds (LOD) score of 2.98 for SNPs between and including rs12142220 to rs490680. This comprised a ~24 Mb linkage region spanning chr1: 84,662,217–108,724,518, hg19 (Supplementary Fig. 1). The LOD score of 2.98 was achieved at rs1801265. Analysis of srWGS from AUS1-V:3 using EHdn identified a

✉ e-mail: andrea.cortese@ucl.ac.uk; gina.ravenscroft@uwa.edu.au

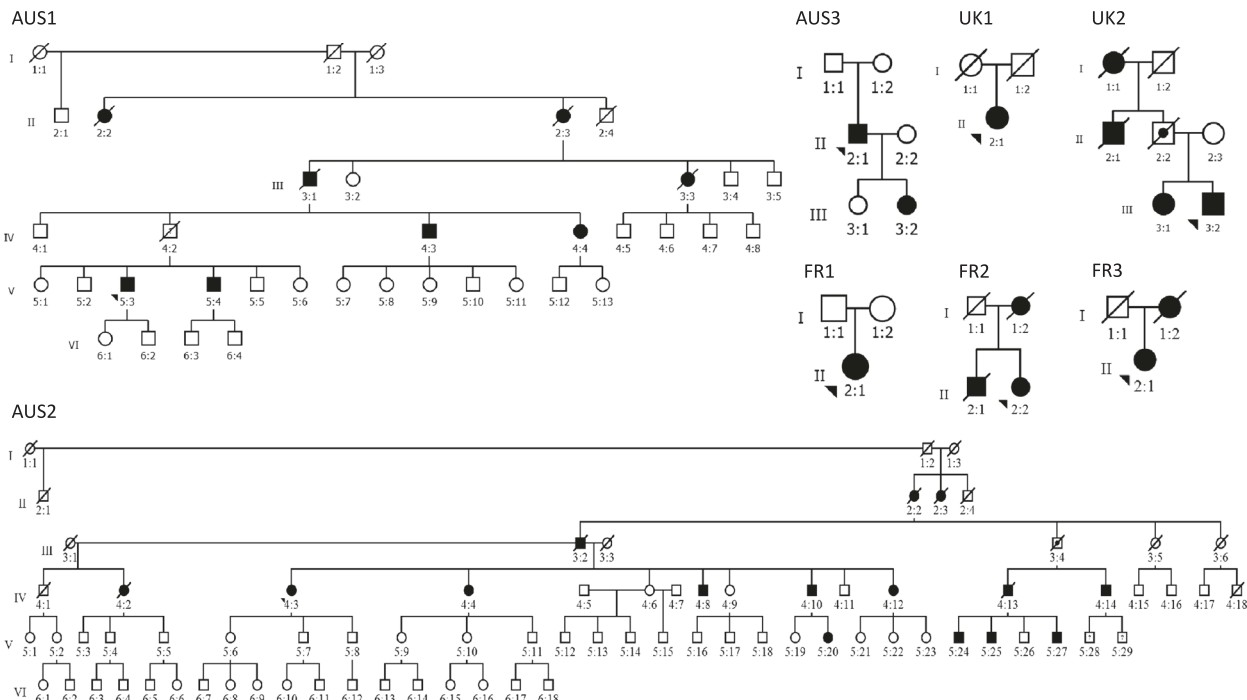

**Fig. 1 | European and Australian OPDM pedigrees.** Pedigrees of the eight OPDM-affected families included in this study. Obligate carriers are indicated and "?" denotes individuals in whom the affection status remains unknown.

CCG repeat expansion within the untranslated first exon of *ABCD3* and called this expansion as 72 repeats. Visual inspection of the BAM file in the Integrated Genome Viewer revealed that the 5'-UTR of *ABCD3* was spanned by >15 reads which confirmed the presence of a CCG repeat expansion (Fig. 2A).

Notably, the *ABCD3* repeat locus was independently identified through an unbiased analysis of the Genomics England 100,000 Genome Project - Rare Disease Cohort, thus strengthening the validity of the finding. Based on the presence of a shared pathogenic expanded CGG • CCG repeat motif in different genes underlying OPDM in cases of East Asian ancestry, we speculated that CGG • CCG expansion in additional genes may cause OPDM in cases of European origin. Therefore, we leveraged srWGS data from 371 cases with myopathy enroled in the 100,000 Genome Project. We specifically looked for loci containing CGG • CCG expansions, which were enriched in the myopathy group compared to controls. The locus with the highest z-score identified by EHdn (outlier method) was a CCG repeat located in the 5'-UTR of *ABCD3*. We next performed a more accurate profiling of the repeat locus using ExpansionHunter v3.2.2 which confirmed the presence of a large monoallelic CCG expansion of *ABCD3* in two cases diagnosed with OPDM (estimates of 109 and 98 repeats) (Fig. 2B).

The CCG expansion in *ABCD3* was confirmed by RP-PCR and segregated with the disease in the family UK2, being present in the affected sister UK2-III:1 but absent in their unaffected mother. Unfortunately, DNAs from additional affected family members (UK2-II:1 and UK2-I:2) and the obligate carrier UK2-II:2 were unavailable for testing, as these individuals were deceased.

We next performed RP-PCR to screen the CCG expansion in *ABCD3* in a cohort of 68 cases diagnosed with cranial or distal limb myopathy or full OPDM. We identified three additional probands of French nationality from unrelated families diagnosed with OPDM (FR1-II:1, FR2-II:1, FR3-II:1) carrying the CCG expansion in *ABCD3* (Fig. 2D). In UK1-II:1, UK2-III:1, UK2-III:2, and FR1-II:1 fresh blood was available for extraction of ultralong fragments. Therefore, these samples were subjected to Bionano optical genome mapping, confirming the presence of the *ABCD3* repeat expansion (Fig. 2C).

The existence of two obligate carriers (AUS2-III:4 and UK2-II:2), who both lived until their old age and did not develop the disease, suggests that the repeat may show incomplete penetrance in rare cases. Unfortunately, DNA was not available for assessing the repeat size.

*ABCD3* repeat expansions were absent from optical genome mapping control datasets comprising 974 alleles and are very rare in large srWGS control datasets. There were no large expansions over 50 repeats, corresponding to the average read length of 150 bp, in 32,884 alleles present on GnomAD. Only one expanded allele out of 6538 (0.015%) was identified in the Broad Centre for Mendelian Genomics and Rare Genomes Project (RGP) cohorts and ten expanded alleles out of 71,498 alleles (0.013%) were identified in the 100,000 Genomes Project. In all control individuals the estimated repeat size was lower compared to OPDM cases. Further details of repeat size distribution in control datasets are available in Supplementary note 1 and Supplementary Fig. 2.

**Accurate sizing and methylation profiling of *ABCD3* CCG repeat expansion through targeted ONT long-read sequencing**

Targeted long-read sequencing (LRS) of the *ABCD3* 5'-UTR was performed on DNA from ten affected and two unaffected individuals from families AUS1-3 and nine affected individuals from families UK1, UK2, FR1, FR2, and FR3. LRS identified pure CCG repeat expansions on a single allele in all affected individuals ($n = 19$) ranging from 118 to 694 repeats (average 283 repeats, SEM: 39.8). In all affected individuals the second (short) allele contained seven CCG repeats, the unaffected relatives harboured two alleles containing seven CCG repeats (Fig. 3A). The expanded alleles remained unmethylated in 17/19 OPDM individuals, while in two affected females harbouring large expansions of the expanded allele was hypermethylated (Supplementary Fig. 3). Anecdotally, FR3-I:2 had the largest expansion of 694 repeats, but age of disease onset was relatively late, in her fifth decade of life, suggesting that the hypermethylation of very large repeat could mitigate the phenotype or result in incomplete penetrance. Unfortunately, muscle tissue or cell lines were not available from the two subjects showing

**Table 1 | Clinical findings in 24 affected individuals from eight families with a molecular diagnosis of OPDM due to CCG repeat expansions in the 5'-UTR of *ABCD3***

| ID | Repeat length | Sex | Age at last follow-up (decade) | Age of onset (decade) | First symptom | Ptosis | Ophthalmoparesis | Facial weakness | Distal LL weakness | Distal UL weakness | Proximal LL weakness | Proximal UL weakness | Dysphagia | Dysarthria | Muscle biopsy: vacuoles | Muscle biopsy: p62 intranuclear inclusions | Intranuclear inclusions on EM | CK levels (IU/L) |
|---|---|---|---|---|---|---|---|---|---|---|---|---|---|---|---|---|---|---|
| AUS1-V:3 | 130 | M | 41-50 | 31-40 | Ptosis | Yes | Yes | No | No | No | No | No | No | No | N/A | N/A | N/A | N/A |
| AUS1-V:4 | 120 | M | 41-50 | 41-50 | Ptosis | Yes | No | No | No | No | No | No | No | No | N/A | N/A | N/A | N/A |
| AUS1-IV:3 | 143 | M | 71-80 | 41-50 | Ptosis | Yes | Yes | Yes | No | No | No | No | Yes | Yes | N/A | N/A | N/A | N/A |
| AUS1-III:3 | N/A | F | 71-80 | Nov-20 | Ptosis | Yes | Yes | Yes | Yes | Yes | Yes | Yes | Yes | Yes | Yes | N/A | N/A | N/A |
| AUS3-II:1 | 230 | M | 61-70 | Nov-20 | Ptosis | Yes | Yes | No | Yes | Yes | No | N/A | Yes | Yes | Yes | None | N/A | 321 |
| AUS3-III:2 | 129 | F | 21-30 | N/A | Ptosis | Yes | No | Yes | No | No | Yes | No | No | No | N/A | N/A | N/A | N/A |
| AUS2-IV:2 | 617 | F | 61-70 | 21-30 | Ptosis | Yes | Yes | N/A | Yes | Yes | N/A | N/A | Yes | Yes | No | None | No | N/A |
| AUS2-V:24 | 129 | M | 21-30 | N/A | ptosis | Yes | N/A | Yes | yes | N/A | No | No | N/A | N/A | N/A | N/A | N/A | N/A |
| AUS2-IV:10 | 208 | M | 51-60 | 21-30 | Ptosis | Yes | Yes | Yes | Yes (mild) | Yes | Yes | Yes | Yes | Yes | N/A | N/A | N/A | 257 |
| AUS2-IV:3 | 324 | F | 61-70 | Nov-20 | Ptosis | Yes | Yes | Yes | Yes | Yes (mild) | Yes | No | Yes | Yes | Yes | None | No | N/A |
| AUS2-IV:13 | N/A | M | 41-50 | 21-30 | Ptosis/unable to jump | Yes | Yes | Yes | Yes | Yes (mild) | No | No | Yes | Yes | N/A | N/A | N/A | 348 |
| AUS2-V:25 | N/A | M | 31-40 | N/A | Ptosis (mild) | Yes | No | No | No | No | No | No | No | No | N/A | N/A | N/A | N/A |
| AUS2-V:27 | N/A | M | 31-40 | 0-10 | Ptosis | Yes | No | Yes | Yes | Yes | No | No | Yes | N/A | N/A | N/A | N/A | N/A |
| AUS2-V:20 | N/A | F | 21-30 | 21-30 | Ptosis | Yes | No | No | No | Yes (mild) | Yes (mild) | No | No | No | N/A | N/A | N/A | N/A |
| AUS2-IV:14 | N/A | M | 51-60 | 31-40 | Dysphagia | Yes | Yes | Yes | Yes | Yes | No | No | Yes | Yes | Yes | None | N/A | N/A |
| AUS2-IV:4 | 381 | F | 61-70 | N/A | Ptosis | Yes | Yes | Yes | Yes | Yes | Yes | N/A | Yes | N/A | Yes | None | No | N/A |
| UK1-II:1 | 381 | F | 51-60 | Nov-20 | Ptosis | Yes | Yes | yes | yes | Yes | Yes | No | Yes | No | N/A | N/A | N/A | 142 |
| UK2-III:1 | 231 | F | 51-60 | 31-40 | Ptosis | Yes | Yes | Yes | Yes | Yes | Yes | No | Yes | No | No | None | N/A | N/A |
| UK2-III:2 | 218 | M | 41-50 | Nov-20 | Ptosis | Yes | Yes | Yes | Yes | Yes | Yes | Yes | Yes | Yes | Yes | None | N/A | 259 |
| FR1-II:1 | 243 | F | 41-50 | Nov-20 | Ptosis | Yes | No | Yes | Yes (severe) | Yes (mild) | Yes | Yes | Yes | Yes | Yes | N/A | No | 233 |
| FR2-II:2 | 560 | F | 61-70 | 21-30 | Ptosis | Yes | Yes | Yes | Yes (severe) | Yes | Yes | Yes | Yes | Yes | Yes | N/A | N/A | 377 |
| FR2-I:2 | 118 | F | died at 81-90 | N/A | N/A | Yes | N/A | N/A | Yes | Yes | Yes | Yes | Yes | Yes | N/A | N/A | N/A |  |
| FR2-II:1 | 300 | M | Died at 61-70 | N/A | N/A | Yes | N/A | N/A | N/A | N/A | No | N/A | Yes | N/A | N/A | N/A | N/A | N/A |
| FR3-II:1 | 233 | F | 31-40 | 21-30 | Ptosis and nasal voice | Yes | Yes | Yes | Yes | Yes | No | No | Yes | Yes | Yes | Yes | N/A | 289 |

*EM electron microscopy, LL lower limb, UL upper limb.*

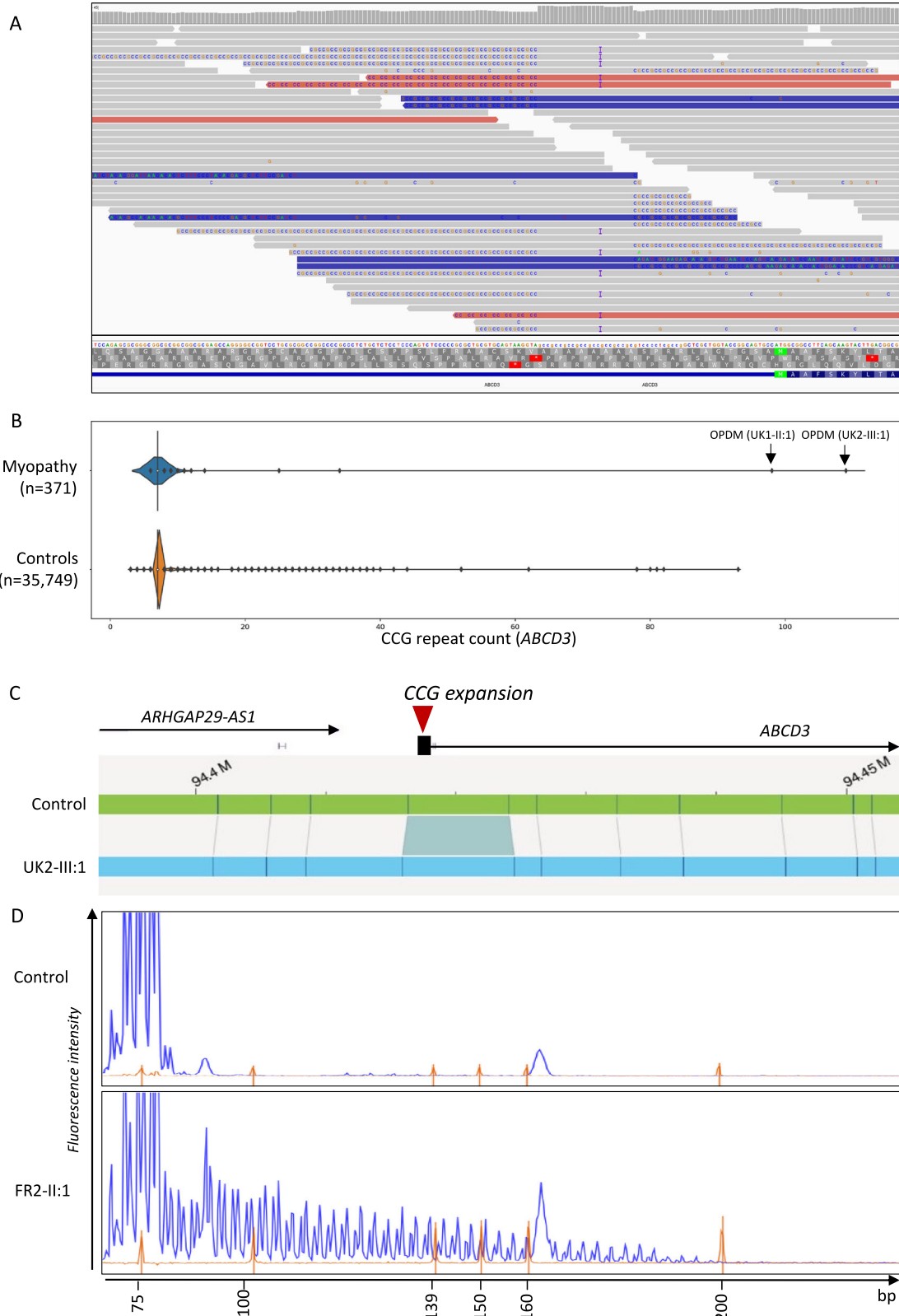

**Fig. 2 | Identification of an expanded CCG repeat within the 5′-UTR of *ABCD3* in OPDM. A** srWGS data (AUS1-V:3) in IGV showing ~50% of reads mapped to the *ABCD3* promoter contain expanded repetitive sequences. **B** Identification of *ABCD3* CCG expansion causing OPDM in srWGS through Genomics England 100,000 Genome Project. **C** Outputs from the Bionano Access® Software showing the presence of an expansion within the 5′-UTR of *ABCD3* in an OPDM individual relative to a healthy control. **D** Electropherogram of repeat-primed PCR. The upper panel showed no-repeat expansion in the unaffected individuals, while the lower panel showed a saw-tooth tail pattern of the repeat expansion in an affected OPDM individual (FR2-II:1).

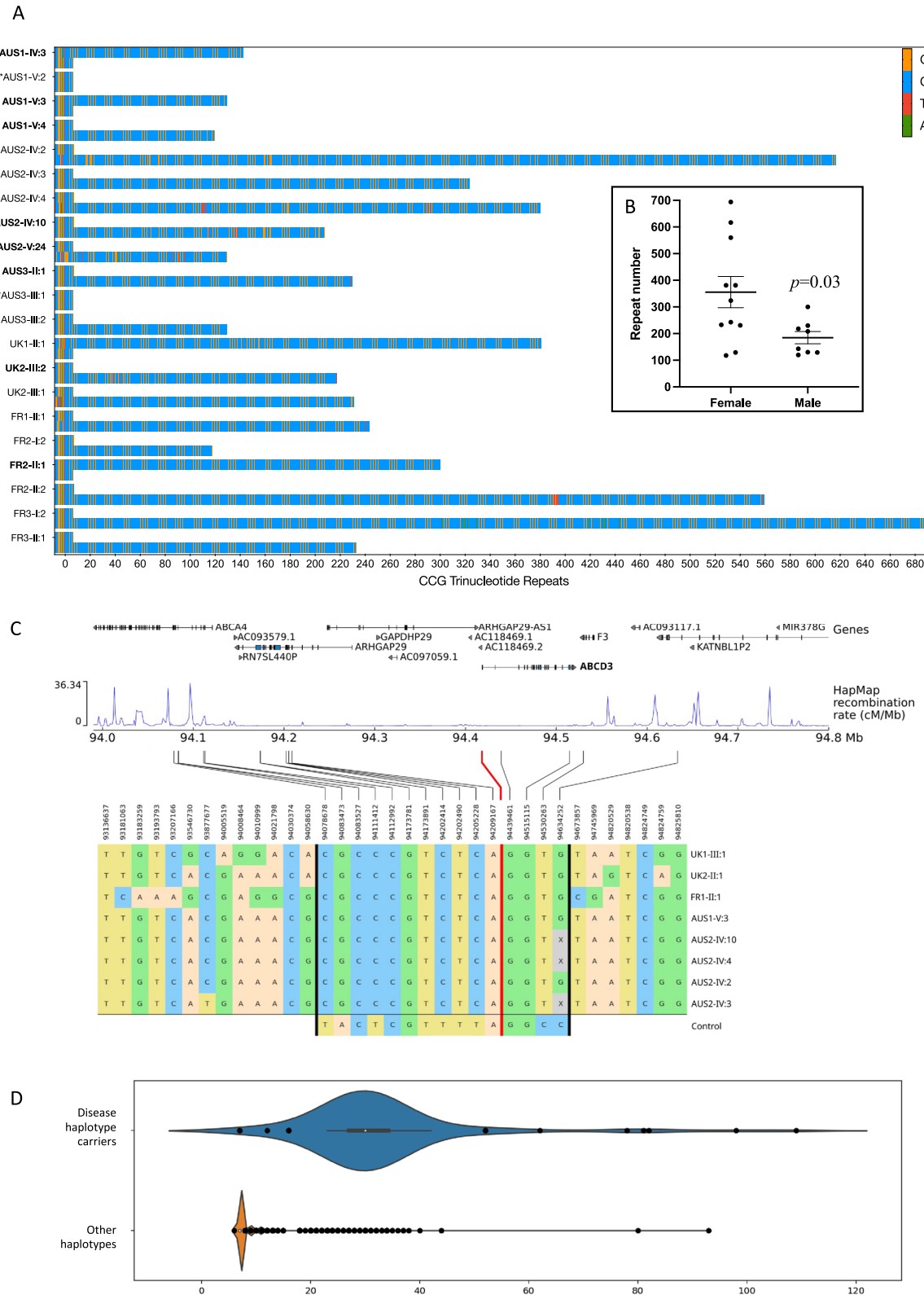

hypermethylated expanded alleles (FR1-II-1 and FR3-I-2) to test this hypothesis.

Notably, EH analysis performed on srWGS data tended to underestimate the actual size of repeat expansions in subjects who also underwent targeted LRS (98 vs 381 repeats in UK1-II:1 and 109 vs 231 in UK2-III:1). This observation confirms previous understanding that, while EH may suggest the presence of large expansions based on

srWGS, it is not reliable for their accurate sizing, particularly when expansions are larger than the sequencing read length (-150 nucleotides).

No repeat expansions at the previously reported loci (*LRP12, GIPC1, NOTCH2NLC, RILPL1*) were identified in any affected individuals, nor were any other alternate candidate repeat expansions identified by EHdn that were shared by the majority of the affected individuals.

**Fig. 3 | Targeted long-read sequencing and analysis of the *ABCD3* repeat expansion. A** Long-read genotyping of the *ABCD3* 5′-UTR STR expansion showed heterozygous pure CCG expansions in all affected individuals (*n* = 19). The wild-type alleles in affected individuals and two unaffected relatives contained 7 x CCG repeats. Unaffected individuals are denoted by an *. Individual IDs for affected males are bolded. **B** Repeat expansions are larger and more variable in size in affected females (*n* = 11) compared to affected males (*n* = 8). Repeat expansion sizes in males vs females were compared with two-sided *t* test and data are presented as mean values ±SEM. **C** The shared haplotype region of 560 kb, identified by 15 SNPs between the vertical black bars. The red line indicates the position of the *ABCD3*

repeat expansion. For comparison, we show in the bottom row the most common haplotype found in controls (18% of alleles). Haplotypes were obtained by phasing genotype data (WGS and WES) with SHAPEITv4. Since the rightmost SNP in the shared haplotype is only available for WGS samples, the four WES samples have been greyed out. **D** Distribution of the repeat count estimation by EH, for carriers of the haplotype (*N* = 96) and non-carriers (*N* = 34,583) in the Rare Disease cohort. Whiskers are based on the standard 1.5 IQR value. For repeat haplotype carriers: min = 7, min whisker = 23, Q1 = 27, median = 30, Q3 = 34, max whisker = 42, max = 109. In non-carriers: min = 6, min whisker = 7, Q1 = 7, median = 7, Q3 = 7, max whisker = 7, max = 93.

## Effect of sex on repeat size and transmission

There was a negative correlation between repeat expansion size and age-of-onset in affected males ($y = 3.029x + 272.8$, $n = 6$, $p = 0.0063$) with larger expansions associated with earlier onset of disease. The repeat expansion size in affected males was $185 \pm 23$ repeats (mean ± SEM, $n = 8$) with a range of 180 repeats; in affected females, the repeats were ~1.9 times longer ($356 \pm 59$ repeats, $n = 11$, $p = 0.0295$) and more variable in length (range 576 repeats) Fig. 3B, suggesting the expansion may be more unstable in females. The onset of the disease in affected females was typically ~5 years earlier than in affected males ($24 \pm 2.9$ vs $29.8 \pm 4.9$) without a significant correlation with repeat expansion size (Supplementary Fig. 4).

Additionally, maternal inheritance of the repeat appears to be associated with incomplete penetrance in the offspring (see pedigrees, Fig. 1). Nine affected/obligate carrier males had 36 children, of which 19 were also affected (penetrance 52.8%), while 11 affected females had 31 children, but only seven were affected (penetrance 22.6%) (Chi square 6.4, df = 1, $p = 0.011$). Also, the two obligate carriers AUS2-III:4 and UK2-II:2 appeared to have inherited the expanded allele from an affected mother.

## Individuals carrying the *ABCD3* CCG repeat expansion share an ancestral haplotype

Subsequently, we looked at the inferred haplotypes associated with *ABCD3* CCG expansions. We identified a region of 450 kb which was shared among all *ABCD3* CCG OPDM samples (14 SNPs between chr1:94078678–94530263), encompassing the entire *ABCD3* gene (Fig. 3C). An additional SNP at position chr1:94634252, available only for samples with srWGS data, extended the shared region by an additional 110 kb downstream of *ABCD3* (total 560 kb). The shared haplotype lies primarily within a low-recombination region (HapMap data) and has an allele frequency of 0.13% in the 100,000 Genome Project (Rare Disease Cohort).

Within the rare disease cohort (number of alleles analysed = 69,358), we analysed the repeat count estimation by ExpansionHunter in individuals carrying the disease-associated haplotype and in non-carriers. Notably, while expanded alleles are observed in both groups, the repeat count was higher in subjects carrying the disease-associated haplotype (median = 30 repeats, IQR = 27–34) vs other haplotypes (median = 7 repeats, IQR = 7–7), with a *p* value of 1.5e-264 ($p < 0.0001$) for the Mann–Whitney test, suggesting that it may represent a more permissive haplotype for the occurrence of large expansions (Fig. 3D).

## Individuals carrying the *ABCD3* CCG repeat expansion present with the hallmark features of OPDM

Comprehensive clinical data from 24 individuals carrying CCG *ABCD3* expansions were available for review (Table 1). The average age of onset was 26.7 years (range: 10–50 years). There were 11 affected males (average age at onset 29.8 years) and 13 affected females (age of onset 24.0 years). In almost all cases the presenting symptom was ptosis (22/24, 92%); one presented with dysphagia and one with weakness of the oropharyngeal muscles. On examination, ptosis was confirmed in all affected individuals, while other common features included

ophthalmoparesis (14/20, 70%), facial weakness (16/21, 77%), dysphagia (16/21, 77%), dysarthria (13/20, 65%) and distal lower limb weakness (17/23, 74%). Figure 4 and Supplementary movie 1 demonstrate the main clinical features and disease progression. Nerve conduction studies and electromyography were in keeping with a myopathic process. CK was normal to mildly elevated ($278 \pm 79$ IU, range 142-377).

Lower limbs muscle MRI was available for one UK proband (UK2-III:2 (Fig. 4L, M). Pattern of involvement was in keeping with the clinical manifestations showing a marked predominant involvement of the distal muscles (mostly soleus and gastrocnemius) compared to the thigh muscles that were mostly preserved.

Skeletal muscle biopsies from nine affected individuals were available for review. Light microscopy revealed the hallmark feature of rimmed vacuoles and other features such as numerous internal nuclei, myofibre splitting, increased fibrous tissue, and internalised capillaries. Rimmed inclusions stained positive for AMP deaminase. Autophagic vacuoles and myeloid bodies were evident on electron microscopy. Rare intranuclear p62-positive inclusions were identified at immunofluorescence only in one muscle biopsy out of five cases (Fig. 5A–I). Muscle biopsies from two cases were available for re-imaging with electron microscopy; careful examination of these biopsies did not identify any intranuclear inclusions (Table 1). It is noted that intra-myonuclear inclusions are only present in ~1% of nuclei in other genetic forms of OPDM[11].

As intranuclear inclusions can often be observed in the skin of individuals carrying CGG • CCG repeat expansions, including neuronal intranuclear inclusion disease, we performed immunohistochemistry in two individuals with OPDM and carrying *ABCD3* expansions. Intranuclear p62-positive inclusions were identified in nuclei of exocrine glands, keratinocytes and fibroblasts in individuals' biopsies (Fig. 5L–N) but were absent or less prominent in controls. Staining of primary fibroblasts generated from skin biopsy of AUS3-IV:3 for p62 found ultra-rare intranuclear p62-positive inclusions (<0.1%, Fig. 5O). Overall, p62-positive nuclei were rare (<1%) in the skin.

## Investigations of disease mechanisms

RNA sequencing and analysis from affected muscle tissue, based on normalised gene counts calculated by OUTRIDER, found that *ABCD3* expression in the three OPDM individuals was higher compared to cases affected by other neuromuscular diseases and healthy controls (Fig. 6A). Skeletal muscle from AUS3-II:1 had the highest expression (2030 [normalised gene count]), followed by AUS2-IV:4 (1582), and AUS2-IV:2 (1404). *ABCD3* was detected as an over-expression outlier in AUS3-II:1 (adjusted *p* value = $1.8 \times 10^{-3}$, log2-fold change = 0.88), but not in AUS2-IV:4, or AUS2-IV:2. Increased expression of *ABCD3* transcript in OPDM muscle tissue was also confirmed by qPCR (Supplementary Fig. 5).

Moreover, no evidence of *ABCD3* antisense transcript was found in our RNA-seq data, nor from public available databases including Cap Analysis of Gene Expression (CAGE) data from the FANTOM5 project[12] (Supplementary Fig. 6).

A key pathological hallmark of repeat expansion disorders is the presence of repeat-containing RNA foci. Therefore, we next performed

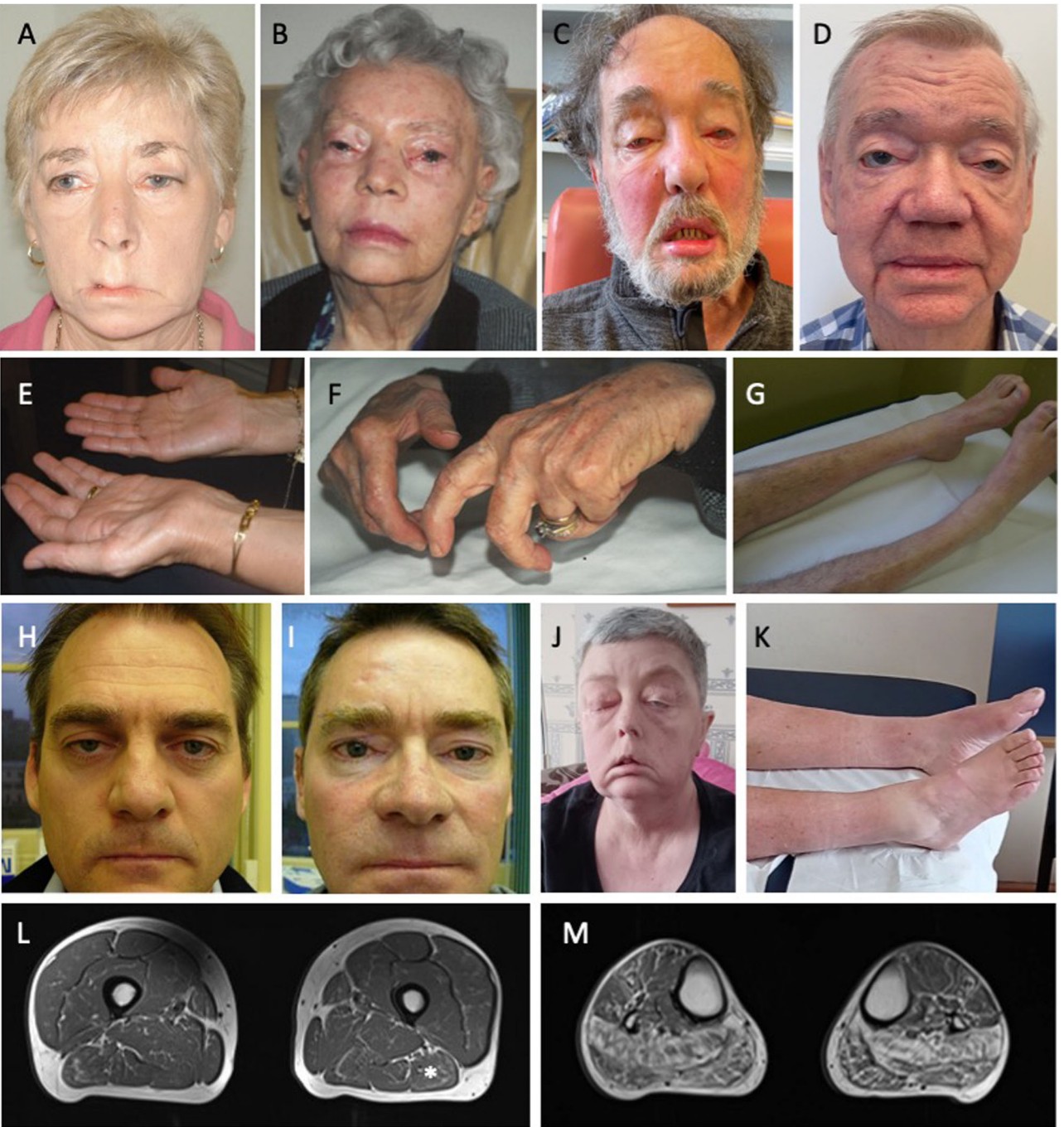

**Fig. 4 | OPDM individuals present with ptosis, and frequently progress to develop weakness of the facial muscles and muscle atrophy and weakness of the distal lower and upper limbs.** Clinical images of AUS2-IV:2 at 67 years of age (**A**), AUS1-II:3 (**B**), AUS3-II:1 at 71 years of age (**C**), AUS1-IV:3 at 77 years of age (**D**) and UK2-III:1 at the age of 52 years of age (**J**) showing ptosis and marked facial weakness. Muscle atrophy of the distal limbs is seen in the hands (**E**) of AUS2-IV:2 (67 years old) and individual AUS1-II:3 developed contractures (**F**). Wasting of the muscles of the lower legs is also seen in AUS3-II:1 (71 years old) (**G**) and UK1-II:1 at 57 years of age (**K**). Mild ptosis is seen in younger affected individuals: AUS1-V:3 at 50 years of age (**H**) and AUS1-V:4 at 48 years of age (**I**). Lower limbs muscle MRI for case UK2-III-2. **L** Thighs are mostly preserved with some mild fatty infiltration predominantly affecting the biceps femoris (\*). **M** Calf muscles are more severely affected. There is a marked predominant involvement of the posterior compartment including soleus and gastrocnemius muscles.

HCR™ RNA-FISH for the *ABCD3* sense transcript on fibroblasts and frozen skin and muscle sections from affected OPDM individuals. We identified increased cytoplasmic and intranuclear signal of *ABCD3* transcript in patient-derived fibroblasts (UK2-III:2) and muscle tissue (AUS2-IV:4) compared to healthy controls (Fig. 6B, C). This is congruent with increased *ABCD3* transcript expression detected in skeletal muscle RNA-seq. In muscle tissue, the *ABCD3* signal appeared clustered in nuclei to form foci like structures. *ABCD3*-positive foci were also identified in the skin biopsy of one OPDM individual but were exceedingly rare (one out of >100 nuclei, Supplementary Fig. 7).

## Discussion

In this study we describe the identification of CCG expansion in 5′-UTR of *ABCD3* causing OPDM in Europeans. A combination of linkage analysis in two large Australian OPDM families (of British origin) and STR analysis of srWGS using EHdn identified the *ABCD3* repeat

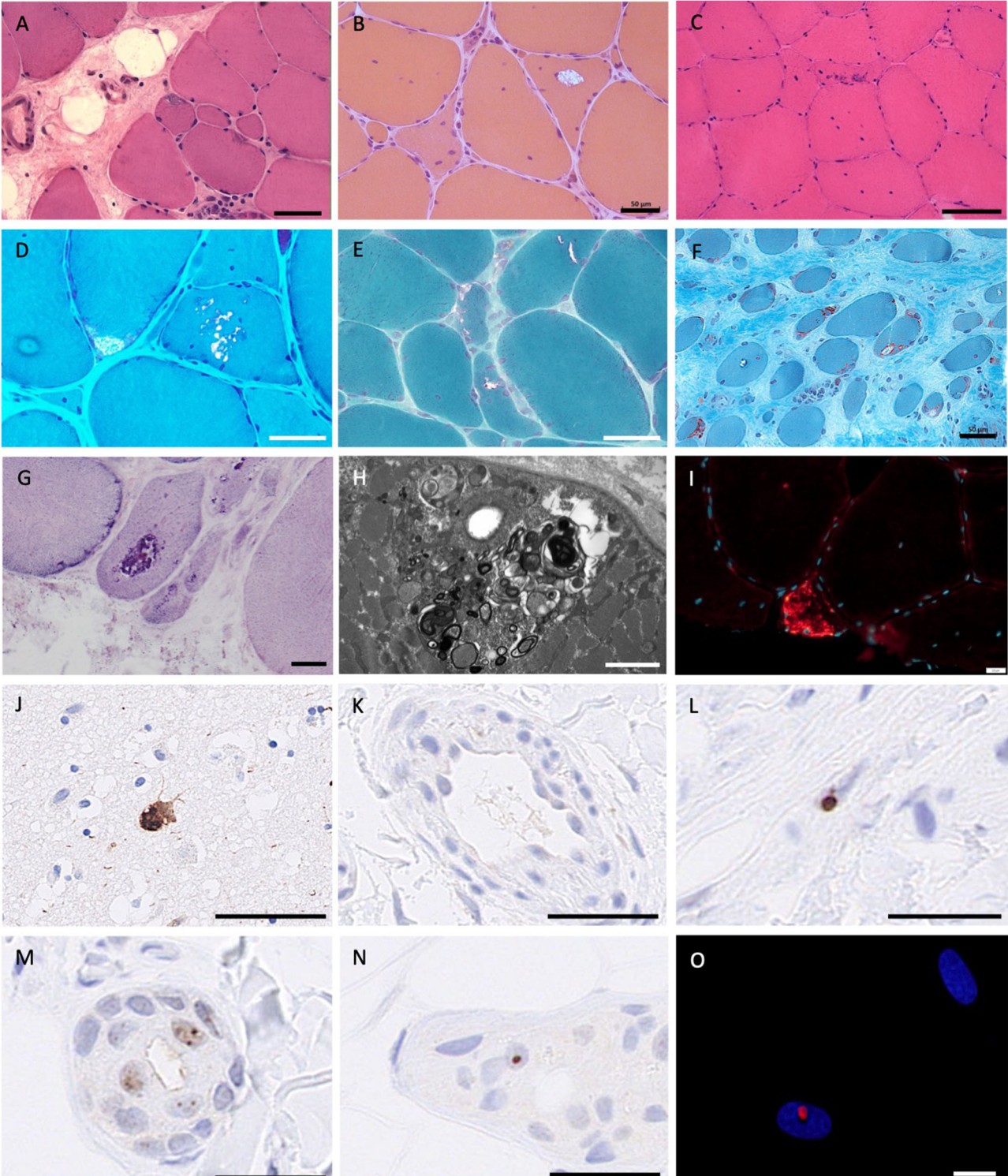

**Fig. 5 | Muscle pathology in *ABCD3*-related OPDM.** Haematoxylin and eosin staining (**A**–**C**) and modified Gomori trichrome staining (**D**–**F**) show rimmed vacuoles, internally localised myonuclei, and variation in myofibre size. Increased fibrotic tissue is also evident. Inclusions also stain with AMP deaminase (**G**). Sub-sarcolemmal myeloid bodies and autophagic vacuoles are evident on electron microscopy (**H**). **I** Strong p62-positivity (red) observed with immunofluorescence in occasional myofibres on a muscle section of the proband from AUS3, AUS3-II:1. No intranuclear p62 inclusions were evident on muscle biopsy. Myonuclei are stained with Hoechst (blue). Individuals: AUS2-IV:3 (**A**), FR3-II:1 (**B**), AUS3-II:1 (**C**), AUS1-II:3 (**D**), AUS1-III:3 (**E**), FR2-II:2 (cricopharyngeal muscle, **F**), AUS2-IV:2 (**G**, **H**). Immunohistochemistry to label p62 (**L**–**N**) showed intranuclear aggregates in fibroblasts (**L**), and exocrine glands (**M, N**) from OPDM individuals UK1-II:1 and UK2-III:2 but not in control skin (**K**). FFPE slide from post-mortem brain of an individual with dementia showing a strong p62 neuronal intranuclear signal used as positive control for the staining (**J**). **O** p62-positive (red) intranuclear inclusion in cultured primary skin fibroblast from an OPDM individual (AUS1-IV:3). Scale bars: 10 μm (**O**), 20 μm (**G, I**), 50 μm (**A, B, D**–**F, K**–**N**), 100 μm (**C, J**). Stainings were carried out once on patients' samples with appropriate controls according to standard practice and histopathology procedures in an ISO15189 accredited laboratory.

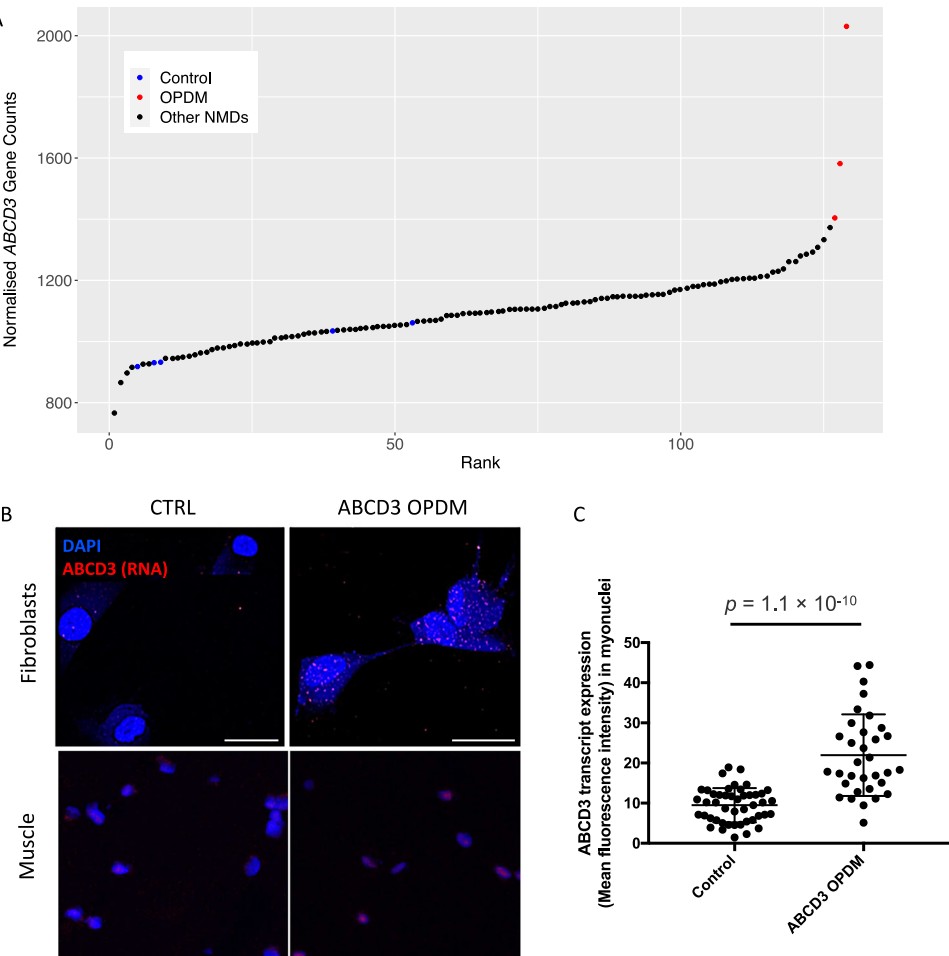

**Fig. 6 | *ABCD3* is overexpressed in OPDM skeletal muscle and forms nuclear and cytoplasmic foci in OPDM-derived fibroblasts and OPDM skeletal muscle.** **A** Analysis of skeletal muscle RNA-seq showed that *ABCD3* appears to be over-expressed in OPDM skeletal muscle compared to healthy controls. The normalised counts were obtained from the OUTRIDER results table. **B** Detection of *ABCD3* transcript in fibroblasts and muscle tissue by HCR RNA FISH. In the OPDM line (top) and muscle (bottom), the intensity of ABCD3 signal (in red) is increased and tends to aggregate in nuclear foci in myonuclei. **C** Increased *ABCD3* nuclear expression in patient myonuclei (*n* = 33) compared to control myonuclei (*n* = 46, *p* < 0.0001). Scale bars: 10 μM. HCR RNA FISH experiments were repeated independently twice with similar results.

expansion as a candidate for disease in these families. Independently, analysis using EHdn on the Genomics England data identified two unrelated individuals within a neuromuscular disease cohort with a clinical diagnosis of OPDM with the same *ABCD3* expansion as outliers. Screening by RP-PCR of additional probands of European ancestry with cranial and distal limb weakness identified four additional unrelated OPDM families (three French and one Australian) with expansions in *ABCD3*.

European individuals with CCG *ABCD3*-related OPDM showed similar features to previously reported East Asian individuals with OPDM (Supplementary Table 1)[6,13]. Onset of the disease was usually in the third decade of life, although the range was broader from the 1st to the 6th decade, and ptosis was usually the presenting symptom. Indeed, upon examination, all individuals had ptosis, while ophthalmoparesis, facial weakness, pharyngeal weakness and distal weakness were present in ~70%. In two families there are two male carriers who lived until old age who did not manifest the disease; unfortunately, we have not been able to size the expanded allele in these individuals. Muscle biopsy often showed mild myopathic changes with rimmed vacuoles, while intranuclear inclusions were exceedingly rare. This aligns with previous reports in East Asian OPDM individuals where intranuclear inclusions were identified in less than 1% of the nuclei in affected muscle tissue[11].

Interestingly, all affected European individuals shared a 560 kb ancestral haplotype encompassing *ABCD3*, which has an allele frequency of 0.13% in the Genomics England 100,000 Genomes Project. There are rare occurrences *ABCD3* repeat expansions (>50 repeats, *n* = 11 alleles) being detected in srWGS data from individuals without OPDM in both the Genomics England and gnomAD datasets (<-0.01%). In two such individuals within Genomics England, for whom follow-up was possible, targeted ONT sequencing found the repeat expansions to be 88 and 120 repeats, below the pathogenic range observed in this OPDM cohort. Of note, Ishiura et al. also found CGG repeat expansions in *LRP12* in a small number of control subjects (0.2%). Also, hexanucleotide repeat expansions in *C9orf72* that cause ALS are seen in 0.15% of UK controls[14] and 0.4% of Finnish controls[15]. Thus, rare occurrences of repeat expansions in healthy individuals are a recurring finding in non-coding repeat expansion diseases, including other genetic forms of OPDM, and suggest the possibility of additional factors, including so far unknown genetic modifiers, contributing to disease penetrance and expressivity.

Our study revealed that *ABCD3* expansion accounted for a relevant proportion of OPDM cases in the Caucasian population, with 15 cases (62%) exhibiting a complete OPDM phenotype. This indicates that *ABCD3* CCG expansions should be considered when investigating individuals with isolated weakness of ocular, facial bulbar or distal.

Notably, over 70 OPDM or OPDM-like European probands tested in this study remain genetically unexplained, indicating the presence of further genetic heterogeneity underlying this condition, even within the European population. While *GIPC1*, *RILPL1*, *NOTCH2NLC* and *LRP12* accounted for ~75% of Asian individuals with OPDM[6], we did not identify any individuals with expansions in these four genes within our European OPDM cohort.

Targeted LRS of 19 *ABCD3* OPDM individuals from eight families found that the pathogenic range for *ABCD3* repeats is 118-694, with repeats being ~1.9 times larger in affected females than males. Although there is no definite explanation for this observation it is possible that the CCG *ABCD3* repeat is prone to further expansion in female embryos. Of note, this phenomenon has been reported in Huntington disease[16,17] and it is hypothesised that X- or Y-chromosome encoded factors involved in DNA replication or repair may influence the repeat stability during the early stages of embryo development[16].

Despite the larger repeat size in females *ABCD3* OPDM individuals were much more likely to inherit the disease from an affected father than an affected mother. We speculate that large maternally inherited *ABCD3* expansions may be prone to further expansion resulting in hypermethylation and silencing of the expanded allele. However, we cannot rule out that very large expansions (>700 repeats) would be deleterious to oocytes or embryonically lethal. Zeng et al. reported that contractions of ultralong *RILPL1* repeats were more common in male-to-offspring transmission than female-to-offspring transmission[6]. This parent-of-origin effect, which is common to several repeat expansion diseases, should be taken into account when counselling affected individuals and their families, as male individuals carrying CCG *ABCD3* repeat expansion are at higher risk of having affected children.

The identification of CGG • CCG repeats as the shared genetic cause of OPDM in functionally different genes suggests a shared pathogenic mechanism underlying myo- and neurodegeneration, at least partly independent of the repeat-containing genes[7,18].

Interestingly, there appears to be a relatively confined pathogenic range associated with CGG • CCG expansions, which is different from most other repeat expansion diseases, reviewed in Malik et al.[19]. Indeed, both small and very large CGG • CCG expansions seem to be tolerated, while disease-causing expansions occur within a specific window.

Boivin et al. determined that the GGC repeat in *NOTCH2NLC* associated with neuronal intranuclear inclusion disease (NIID) and OPDM3 is embedded within a small upstream open reading frame (uORF) and that, via canonical initiation at an AUG start codon, it encodes a small protein (designated uN2C) comprising five amino acids at the N-terminus, a variable central glycine stretch and 38 amino acids at the C-terminus[17]. Using bespoke antibodies targeting the C-terminus of uN2C, Boivin et al. showed that uN2CpolyG is present in the p62-positive intranuclear inclusions seen in NIID patient brain and skin. Over-expression of uN2CpolyG in mice resulted in neuronal toxicity, impaired locomotor activity and reduced lifespan. Thus, confirming that *NOTCH2NLC* GGC repeats result in translation of a polyG containing protein that forms intranuclear inclusions that results in pathology in cells and mice.

Translation of an uORF containing GGC repeat expansions into toxic polyG protein is reminiscent of another neurodegenerative disorder, Fragile X-associated tremor/ataxia syndrome (FXTAS), caused by expansions of 80-200 CGG repeats in the 5'-UTR of *FMR1*, reviewed in Zhou et al.[20]. Studies have shown that the FMRpolyG protein translated via initiation at an ACG near-cognate start codon, like uN2CpolyG, produced from these expansions forms p62-positive intranuclear inclusions and causes toxicity to cells and animal models[21,22].

Although the mechanism underlying muscle degeneration in *ABCD3* repeat expansion remains unknown, RNA-sequencing and qPCR found that *ABCD3* is overexpressed in skeletal muscle from *ABCD3* OPDM individuals. HCR RNA-FISH for *ABCD3* sense transcripts also confirmed an increased transcript abundance in patients' fibroblasts and muscle tissue compared to controls, with clusters of signal in some nuclei suggestive of RNA foci. These foci are thought to represent repeat-containing RNAs and specific RNA-associated binding proteins, which could be detrimental to myofibre survival, a mechanism that is well documented in myotonic dystrophy[23] and suggested in another OPDM subtype[6,7]. Potential RNA toxicity mechanisms[19] have also been proposed in *RILPL1*-related OPDM[7] Alternatively, or concurrently, repeat peptides which are synthesised through repeat associated non-ATG dependent (RAN) translation[19], as shown in *C9orf72*-related ALS and repeat associated translation from the *NOTCH2NLC* 5'-UTR, could contribute to the progressive muscle degeneration observed in *ABCD3*-mediated disease. Using a recently developed in silico tool[24] we predicted that the most likely amino-acid stretches produced from RAN translation of the *ABCD3* repeat expansion are alanine (sense) and alanine or glycine (antisense). Since transcriptomic data suggests *ABCD3* antisense transcripts are not present or lowly abundant in skeletal muscle from patients and controls it is tempting to speculate that poly-Ala stretches may be preferentially generated at this locus (Supplementary Fig. 8). Experimental determination of the contribution of RAN-mediated translation to the pathobiology of *ABCD3* repeated OPDM is beyond the scope of this initial study.

In summary, we describe here the identification of CCG repeats in the 5'-UTR of *ABCD3* as a cause of OPDM in families of European ancestry and show that all affected individuals share a common ancestral haplotype. This study improves our understanding of the molecular aetiology of OPDM and suggests that CCG and CGG expansions in 5'-UTRs of other genes may explain additional cases of OPDM.

## Methods
### Patients
This research complies with all relevant ethical regulations. This study was approved by the Human Research Ethics Committee of the University of Western Australia (RA/4/20/1008), the Human Research Ethics Committee of the Royal Children's Hospital (HREC 28097) and Northeast-Newcastle & North Tyneside 1 Research Ethics Committee (22/NE/0080). All individuals provided informed consent and we obtained consent from the affected individuals or their relatives (in the case of deceased individuals) to share clinical images and video. Supplementary Movie 1 shows a brief clinical examination and presentation of features in *ABCD3*-related OPDM. The authors affirm that human research participants provided informed consent for publication of the images in Fig. 4 and Supplementary Movie 1. Also, ethical approval and informed consent were obtained specifically for derivation and use of human skin fibroblasts, and for collection and use of muscle biopsies in this research study (from all individuals, including controls).

Patients diagnosed clinically with OPDM and genetically unconfirmed myopathy were recruited from participating centres in Australia, the United Kingdom (UK), France and through a network of collaborators worldwide (OPDM study group). The following information was collected for all patients using a standardised template: demographics, family history, current age, age at disease onset, first symptom/s, presence of ptosis, ophthalmoparesis, facial weakness, dysphagia, dysarthria, distribution of limb weakness, respiratory involvement, age at death. When available, EMG were reviewed.

Muscle biopsies were performed on nine individuals and routine histopathology conducted as part of the clinical work-up. Investigations included H&E, Gomori trichrome and NADH staining. P62 staining was also available in X cases. Ultrastructural studies were performed following standard methods[25]. A small fragment of the

muscle was fixed in 2.5% w/v glutaraldehyde solution, postfixed in 1% w/v osmium tetroxide, and embedded in epoxy resin. Semithin sections were stained with 1% toluidine blue. Ultrathin sections were mounted on copper grids and examined with a Zeiss Libra 120 transmission electron microscope.

### Linkage
Linkage was performed using MERLIN[26] and whole exome sequencing data (WES) on five affected individuals from AUS1 (III:3, IV:3, IV:4, V:3, V:4) and four affected individuals from AUS2 (IV:2, IV:3, IV:4, IV:10). The following parameters, which are standard for autosomal dominant disease were used: Disease allele frequency: 0.01, Penetrance (Probability of being affected with): 0 alleles: 0.001, 1 allele: 1.0, and 2 alleles: 1.0.

### Short read (sr)-WES and srWGS and analysis in Australian families
Ampliseq srWES was performed on DNA from affected individuals in the Australian families (AUS1, AUS2) as described previously[27].

PCR free srWGS (2 × 150 bp reads, >30× coverage) on the proband in Family AUS1 (V:3) was performed at The Centre of Mendelian Genomics, The Broad Institute. ExpansionHunter denovo (EHdn)[28] was used to identify repeat expansions within the linkage region.

### srWGS analysis in the Genomics England 100,000 genome project
The 100,000 Genomes Project, run by Genomics England, was established to sequence whole genomes of patients of the National Health Service (NHS) of the UK, affected by rare diseases and cancer[29,30]. We analysed a cohort of 371 myopathy cases and 35,749 non-neurological controls enroled in the 100,000 Genome Project - Rare Disease Cohort. For all individuals srWGS was available (paired-ended, 150 bp, average read depth >30×). All samples were profiled with EHdn v0.9.0 to obtain a count of anchored in-repeat reads, then jointly analysed using the "outlier" method provided in the EHdn package. The method computes a z-score for each sample, based on the original distribution of read counts. The score is to be interpreted as distance to the median, relative to the distribution width.

### ABCD3 haplotype analysis
We used SHAPEITv4 with default parameters to phase a 3 Mb region (chr1:92500000–95500000 Hg38) encompassing ABCD3. To maximise available haplotype information, the entire Rare Diseases cohort of Genomics England (78,195 samples from patients affected by rare diseases, including the OPDM cases UK1-II:1 and UK2-III:2) was jointly phased. Input data format was an aggregate VCF file with a total of 599,962 variants. External OPDM cases (AUS1-V:3, AUS2-IV:2, AUS2-IV:3, AUS2-IV:4, AUS2-IV:10, FR1-II:1) were subsequently phased using the previously phased data as reference.

### Repeat-primed PCR (RP-PCR)
RP-PCR was performed to study segregation of CCG expansion in families UK1 and UK2 and to screen ABCD3 CCG expansion in a cohort of 68 cases diagnosed with either OPDM ($n = 28$), chronic progressive external ophthalmoplegia ($n = 25$) either isolated or as part of more widespread myopathic involvement, or distal myopathy ($n = 15$). The following primers were used: ABCD3 Fw: FAM-GCTCTCCTCCCAGTCTCCCC; anchor: CAGGAAACAGCTATGACC; ABCD3 Rv: CAGGAAACAGCTATGACCCGGCGGCGGCGG. The PCR mix contained 0.5 U Takara LaTaq (Takara, Shiga, Japan), 1× GC Buffer, 400 μM each dNTP mixture, 0.4 μM primer ABCD3-Fw, anchor and 0.04 μM ABCD3-Rv, <1 μg DNA, and ddH2O for a final volume of 50 μL. We used the following thermal conditions: pre-denaturation (95 °C for 5 minutes), followed by 50 cycles of 95 °C for 30 seconds, 62 °C for

1 minute, 72 °C for 2 minutes and final extension (72 °C for 5 minutes). The ramp rate to 95 °C and 72 °C was set to 2.3 °C s⁻¹ and that to 62 °C was set to 1.5 °C s⁻¹. Fragment analysis was conducted on an ABI-3730XL Sequencer, and data were visualised using Geneious Prime (Biomatters Ltd).

### Analysis of the ABCD3 locus in srWGS from population datasets
To examine the population distribution of STR alleles at the ABCD3 locus, we ran ExpansionHunter on 16,442 srWGS samples from gnomAD by specifying the locus region as chr1:94418421-94418442 (GRCh38, 0-based coordinates) and repeat unit as GCC. We also used ExpansionHunter to genotype 3270 srWGS rare disease samples from the RGP and the Centre for Mendelian Genomics (CMG). These included 1608 cases from a wide range of disease and phenotype categories and 1662 unaffected family members.

Finally, we profiled the ABCD3 CCG expansion locus with EH in the entire 100,000 Genome Project cohort of 35,749 srWGS from non-neurological controls from the UK. We analysed its size in 724 controls, and 250 alleles from internal non-OPDM samples that underwent optical genome mapping at UCL Institute of Neurology (see methods below).

### Programmable targeted nanopore sequencing and analysis
High molecular weight (HMW) DNA samples were transferred to the Garvan Institute's Sequencing Platform for LRS analysis on Oxford Nanopore Technologies (ONT) instruments. Prior to ONT library preparations, the DNA was sheared to ~20 kb fragment size using Covaris G-tubes and visualised, post-shearing, on an Agilent TapeStation. Sequencing libraries were prepared from ~3-5ug of HMW DNA, using native library prep kit SQK-LSK110, according to the manufacturer's instructions. Each library was loaded onto a FLO-MIN106D (R9.4.1) flow cell and run on an ONT MinION device with live target selection/rejection executed by the ReadFish software package[31] Detailed descriptions of software and hardware configurations used for Read-Fish experiments are provided in a recent publication that demonstrates the suitability of this approach for profiling tandem repeats[32]. Samples were run for a maximum duration of 72 hours, with nuclease flushes and library reloading performed at ~24- and 48-hour timepoints for targeted sequencing runs, to maximise sequencing yield.

Raw ONT sequencing data was converted to BLOW5 format[33] using slow5tools (v0.3.0)[33] then base-called using Guppy (v6). Resulting FASTQ files were aligned to the hg38 reference genome using minimap2 (v2.14-r883)[34]. The short-tandem repeat site within ABCD3 was genotyped using a process validated in our recent manuscript[32]. This method involves local haplotype-aware assembly of ONT reads spanning a given STR site and annotation of STR size, motif and other summary statistics using Tandem Repeats Finder[35] followed by manual inspection and motif counting. DNA methylation profiling was performed with F5C (v1.1)[33].

Targeted ONT sequencing of the four previously described OPDM loci was also performed on affected individuals from each family.

### Bionano optical genomic mapping
Patients UK1-II:1, UK2-III:1, UK2-III:2, UK1-II:1 and FR1-II:1 for whom whole blood was available were subjected to Bionano optical mapping to gather additional information on the precise size of the expanded repeat. HMW genomic DNA was isolated using kits provided by Bionano Genomics, as described in Bionano Prep SP Frozen Human Blood DNA Isolation Protocol v2. Homogeneous HMW DNA was labelled using Bionano Prep Direct Label and Stain Protocol with kit provided, and the homogeneous labelled DNA was loaded onto a Saphyr chip. Optical mapping was performed at theoretical coverage of ×400. Molecule files (.bnx) were then aligned to hg38 with Bionano Solve script "align_bnx_to_cmap.py" using standard parameters.

## Skeletal muscle RNA-seq

RNA was extracted from patient and control skeletal muscle biopsies (~15–50 mg) using the RNeasy Fibrous Tissue Mini Kit (QIAGEN #74704). RNA purity was assessed by Nanodrop, followed by PCR amplification and 1% agarose gel electrophoresis to confirm the absence of DNA. Sample sequencing was performed at Genomics WA (Harry Perkins Institute of Medical Research, Perth, Australia). Stranded Poly A RNA-seq libraries were prepared from extracted RNA using Agilent Sureselect XT library preparation kit. The protocol includes poly A enrichment followed by fragmentation, reverse transcription, ligation with adaptors and amplification for indexing. QC was performed using TapeStation 4200 and Qubit, and QC sequencing on the Illumina iSeq 100 flow cell. Paired-end sequencing (strand-specific reverse) was performed on the Illumina NovaSeq 6000 instrument using a $2 \times 150$ cycle configuration to a depth of 50 million read pairs per sample. Adaptor sequences were removed and demultiplexed FASTQ files were provided by Genomics WA for download and further analysis.

Processing of FASTQ files, including read quality control and alignment, was performed using the nf-core/rnaseq pipeline (https://nf-co.re/rnaseq/1.3), version 3.8.1. Raw reads were aligned to the GRCh38 human reference genome using STAR (version 2.7.10a). Detection of aberrant expression was achieved using DROP (version 1.3.3)[36], as previously described[37]. DROP leverages OUTRIDER, a method for detecting aberrant RNA-seq read counts, which uses a denoising autoencoder to control for latent effects and returns multiple-testing corrected $p$ values (FDR) for each gene and sample. The cohort consisted of 130 skeletal muscle RNA-seq datasets from rare neuromuscular disease patients, including three OPDM patients and five unaffected controls.

## Quantitative PCR for ABCD3 expression

qPCRs were performed to determine the relative expression of *ABCD3* in skeletal muscle available from three patients (AUS2-IV:4, AUS2-IV:2, AUS3-II:1) and three healthy controls. RNA was isolated as described above and cDNA was synthesised using the LunaScript® RT SuperMix Kit (NEB, Cat#E3010) from 200 ng RNA input. qPCR reactions were performed using the QuantiNova SYBR Green PCR Kit (QIAGEN, Cat#208056) and consisted of 1× QuantiNova SYBR Green PCR master mix, QN ROX Reference Dye (0.05 μl per reaction), 1 μl cDNA (5 ng) and 0.7 μM each of forward and reverse primers in a total volume of 10 μl. Each reaction was performed in duplicate. The QuantStudio™ 6 Pro Real-Time PCR System and the Thermo Fisher Connect platform were used for thermal cycling and data analysis, respectively. Cycling conditions were: 95 °C/2 min, (95 °C/5 s, 60 °C/17 s) ×40, followed by melt curve analysis. Transcript abundance was normalised to the geometric mean of two reference genes (*TBP* and *EEF2*) using the delta Ct method. Reported values are the mean ± SD.

## Fibroblasts cultures

Human derived skin fibroblasts from OPDM patients (UK2-III:1 and AUS1-IV:3) and age-matched controls were cultured in Dulbecco's Modified Eagle Media supplemented with 10% FBS at 37 °C in a 5% $CO_2$ humidified incubator.

## Immunofluorescence and imaging

For detection of p62 aggregates in skin, formalin-fixed paraffin-embedded (FFPE) slides were stained with the mouse monoclonal antibody against p62 (1:100 dilution, 3/P62LCK Ligand, BD Transduction) and developed using the OptiView DAB IHC Detection Kit (Roche Diagnostics).

Primary fibroblast cultures from AUS1-IV:3 were grown in 24 well tissue culture plates, fixed and permeabilised with 2% paraformaldehyde and 1% saponin in phosphate buffered saline (PBS) for 15 minutes. Cells were then blocked in PBS containing 10% fetal calf serum, 5% goat serum and 1% bovine serum albumin (blocking buffer) for 60 minutes at room temperature. Cells were then incubated with a mouse monoclonal antibody against p62 (1:50 dilution, Abcam, ab56416) overnight at 4 °C. Cells were washed 3 × 5 minutes with PBS and then incubated with goat anti-mouse IgG2a AlexaFluor® 555 (1:500) for 60 minutes at room temperature in blocking buffer. Cells were washed in PBS and counterstained with Hoechst (Sigma, Australia). Imaging was performed on an inverted fluorescent microscope (model IX-71, Olympus) with a digital camera (model DP-74, Olympus).

## Hybridisation chain reaction (HCR) RNA fluorescence in situ hybridisation (FISH)

We performed RNA in situ hybridisation for *ABCD3* transcript on human derived skin fibroblasts from one patient (UK2-III:1) and three age-matched controls; and on frozen muscle sections from one individual (AUS2-IV:4) and one healthy control. HCR™RNA-FISH significantly amplifies the signal of an individual molecule over traditional FISH and was shown to be more sensitive than FISH at detecting low abundant RNAs and endogenous G4C2 repeats in *C9orf72* ALS-FTD patient brains[38]. HCR™RNA-FISH (Molecular Instruments) was performed on fibroblasts according to manufacturer's protocol. Briefly, $4 \times 10^5$ cells were seeded on coverslips and fixed in 4% formaldehyde for 10 minutes at room temperature. Cells were then washed twice in PBS and permeabilised in 70% cold ethanol overnight at −20 °C. After two washes in 2× saline sodium citrate (SSC), cells were pre-warmed in probe hybridisation buffer (Molecular Instruments) at 37 °C for 30 minutes and then incubated with 1.2 pmol of *ABCD3* probe set in hybridisation buffer at 37 °C overnight. The following day cells were washed four times with probe wash buffer (Molecular Instruments) at 37 °C and twice with 5× SSC + 0.1% Tween-20 at room temperature (5× SSC-T). A pre-amplification step was performed with probe amplification buffer (Molecular Instruments) for 30 minutes at room temperature and cells were incubated with 18 pmol hairpins B1H1 and B1H2 at room temperature overnight. Cells were washed 5× SSC-T at room temperature, incubated with DAPI for 15 minutes and then mounted with Dako mounting medium.

For frozen muscle sections, samples were processed according to Molecular Instrument's protocol. Briefly, slides were fixed in 4% paraformaldehyde for 15 minutes at 4 °C and subsequently dehydrated with a graded series of ethanol. After two washes in PBS, sections were pre-heated in probe hybridisation buffer (Molecular Instruments) for 10 minutes at 37 °C and then incubated with 1.6 pmol of *ABCD3* probe set in hybridisation buffer at 37 °C overnight. The following day slides were washed in four times in probe wash buffer and gradually increasing concentrations of 5× SSC-T at 37 °C and once in 5× SSC-t at room temperature. The amplification stage was performed as described above for the skin fibroblasts. After three washes in 5× SSC-T at room temperature, slides were treated with 0.1% Sudan Black in ethanol 70% and washed thrice in ethanol 30%. Finally, slides were counterstained with DAPI for 15 minutes and then mounted with Dako mounting medium.

Images were taken on a Zeiss LSM 710 confocal microscope at ×40 magnification. The signal was normalised to control samples.

## Statistical analyses

Data on *ABCD3* repeat length are presented as mean ± standard error of the mean. A two-tailed $t$ test was conducted to determine whether there was a significant difference between *ABCD3* repeat expansion size in affected females compared to affected males, with a threshold for significance set at $p < 0.05$. To compare the proportion of affected children born to affected females vs affected males a Chi square test was performed, with significance set at $p < 0.05$. $T$ test was also performed to compare *ABCD3* transcript expression level in patients vs controls by qPCR and FISH.

**Reporting summary**

Further information on research design is available in the Nature Portfolio Reporting Summary linked to this article.

## Data availability

Human research participants' genotyping microarray and DNA sequencing data generated in this study cannot be deposited or shared on request because the individuals did not provide consent for data deposition/sharing. DNA sequencing data from control individuals and myopathy cases from the 100,000 Genome Project used in this study are accessible and requests should be addressed to Genomics England Limited (https://www.genomicsengland.co.uk/research/academic/join-research-network) DNA sequencing data from control individuals for the Broad CMG cohort is available via dbGaP accession number phs001272. Access is managed by a data access committee designated by dbGaP and is based on intended use of the requester and allowed use of the data submitter as defined by consent codes. The gnomAD data are in part based on data that are available in TOPMed or dbGaP including[1]: data generated by The Cancer Genome Atlas (TCGA) managed by the NCI and NHGRI (accession: phs000178.v10.p8); information about TCGA can be found at http://cancergenome.nih.gov[2]; data generated by the Genotype-Tissue Expression Project (GTEx) managed by the NIH Common Fund and NHGRI (accession: phs000424.v7.p2)[3]; data generated by the Alzheimer's Disease Sequencing Project (ADSP), managed by the NIA and NHGRI (accession: phs000572.v7.p4). For a full list of projects included in gnomAD, please see https://gnomad.broadinstitute.org/about. Aggregated gnomAD STR genotype data is displayed on the browser https://gnomad.broadinstitute.org/, and is available for download on https://gnomad.broadinstitute.org/downloads through Google Cloud Public Datasets, the Registry of Open Data on AWS, and Azure Open Datasets. RNA-seq data generated in this study are available in the SRA database under accession no. PRJNA1117762. All other data generated in this study are available in the article and its Supplementary Information files.

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

## Acknowledgements

We thank professor Ichizo Nishino, Carolin Sewry, and Werner Stenzel for reading and review of the muscle biopsies and electron microscopy. We thank Ms Audrey Rick for assistance with drawing the pedigrees. This research was made possible through access to data in the National Genomic Research Library, which is managed by Genomics England Limited (a wholly owned company of the Department of Health and Social Care). The National Genomic Research Library holds data provided by patients and collected by the NHS as part of their care and data collected as part of their participation in research. The National Genomic Research Library is funded by the National Institute for Health Research and NHS England. The Wellcome Trust, Cancer Research UK and the Medical Research Council have also funded research infrastructure. This project was supported by an NHMRC Ideas Grant (APP2002640) to G.R., N.G.L., M.R.D., P.J.Lamont and M.C-S and Medical Research Council (MR/T001712/1), Fondazione Cariplo (grant n. 2019-1836), the Inherited Neuropathy Consortium, Fondazione Regionale per la Ricerca Biomedica (Regione Lombardia, project ID 1751723) and Italian Ministry of Health (Ricerca Corrente 2021-2022) to A.C. C.K.S. and L.D. are supported by an Australian Government Research Training Programme (RTP) Scholarship, I.W.D. is supported by an MRFF Investigator Grant (MRF1173594) and G.R. is supported by an NHMRC EL2 Investigator Grant (APP2007769). M.B. is supported by an NHMRC L1 Investigator Grant (APP1195236), This work was also supported by the Australian State of Victoria's Government's Operational Infrastructure Support Programme, and the NHMRC Independent Research Institute Infrastructure Support Scheme (IRIISS). This work was supported by resources provided by the Pawsey Supercomputing Research Centre with funding from the Australian Government and the Government of Western Australia. Library preparation and RNA-sequencing was conducted in the Genomics WA Laboratory in Perth, Australia. This facility is supported by BioPlatforms Australia, State Government Western Australia, Australian Cancer Research Foundation, Cancer Research Trust, Harry Perkins Institute of Medical Research, Telethon Kids Institute and the University of Western Australia. We gratefully acknowledge the Australian Cancer Research Foundation and the Centre for Advanced Cancer Genomics for making available Illumina Sequencers for the use of Genomics WA.

## Author contributions

The work was conceived and directed by A.C. and G.R., who wrote the paper with input from all authors. S.J.B., S.F., I.W.D., G.M., C.F., P.J.Lo. performed bioinformatic analyses. R.C. performed the in situ hybridisation staining. E.B. and M.C.-S. had a major role in the collection and analysis of patients' clinical data. I.S., S.R.C., H.G., B.W., C.F., G.M., C.K.S., L.D., M.J., B.R.G., M.E., L.G.F., R.T., J.R., A.M., N.D., E.V., R.P.S., G.F.-E., M.M., D.G., M.B.D., E.S., M.G., D.J.A., G.N., S.V., R.D.H., T.R., J.D., V.F., F.M., M.R.D., M.K., R.Q., S.H., A.T., M.B., C.A.M., N.G.L., T.S., H.H., M.G.H., P.J.Lo., P.J.La., M.C.F. contributed to clinical and pathology data collection, experimental work, and analyses and critically revised the manuscript.

## Competing interests

H.G. has previously received travel and accommodation expenses from ONT to speak at conferences. H.G. and I.W.D. have paid consultant roles with Sequin PTY LTD. M.C.F. has paid consultancy roles with Fenix Innovations, the Victorian Government and PTC. The remaining authors declare no competing interests.

## Additional information

Andrea Cortese ⬚[1,2,61] ✉, Sarah J. Beecroft ⬚[3,61], Stefano Facchini[1,2], Riccardo Curro[1,2], Macarena Cabrera-Serrano ⬚[4,5], Igor Stevanovski ⬚[6,7], Sanjog R. Chintalaphani[6,7], Hasindu Gamaarachchi[6,7,8], Ben Weisburd ⬚[9], Chiara Folland[4,10], Gavin Monahan ⬚[4,10], Carolin K. Scriba ⬚[4], Lein Dofash[4,10], Mridul Johari[4,10], Bianca R. Grosz[11,12], Melina Ellis[11,12],

Liam G. Fearnley [13,14], Rick Tankard [15], Justin Read[16,17], Ashirwad Merve[18], Natalia Dominik[1], Elisa Vegezzi[19], Ricardo P. Schnekenberg[1,2], Gorka Fernandez-Eulate[20], Marion Masingue[20], Diane Giovannini[21], Martin B. Delatycki[16,17], Elsdon Storey[22], Mac Gardner[23], David J. Amor [16,17], Garth Nicholson [11,24], Steve Vucic [12,25], Robert D. Henderson[26,27], Thomas Robertson[28,29], Jason Dyke [30,31], Vicki Fabian[30], Frank Mastaglia[32], Mark R. Davis[33], Marina Kennerson[11,12,24], OPDM study group*, Ros Quinlivan[34], Simon Hammans[35], Arianna Tucci[1,36], Melanie Bahlo [13,14], Catriona A. McLean [37,38], Nigel G. Laing[4,10], Tanya Stojkovic[20], Henry Houlden [1], Michael G. Hanna[1], Ira W. Deveson [6,7], Paul J. Lockhart [16,17], Phillipa J. Lamont[39], Michael C. Fahey [40], Enrico Bugiardini [1,62] & Gianina Ravenscroft [4,10,62] ✉

[1]Department of Neuromuscular Diseases, UCL Queen Square Institute of Neurology, London, UK. [2]Department of Brain and Behavioral Sciences, University of Pavia, Pavia, Italy. [3]Pawsey Supercomputing Research Centre, Kensington, WA, Australia. [4]Harry Perkins Institute of Medical Research, Nedlands, WA, Australia. [5]Department of Neurology and Instituto de Biomedicina de Sevilla, Hospital Universitario Virgen del Rocío/Universidad de Sevilla/CSIC, Sevilla 41013, Spain. [6]Genomics and Inherited Disease Program, Garvan Institute of Medical Research, Sydney, NSW, Australia. [7]Centre for Population Genomics, Garvan Institute of Medical Research and Murdoch Children's Research Institute, Sydney, NSW, Australia. [8]School of Computer Science and Engineering, University of New South Wales, Sydney, NSW, Australia. [9]Program in Medical and Population Genetics, Broad Institute of MIT and Harvard, Cambridge, MA, USA. [10]Centre for Medical Research, University of Western Australia, Nedlands, WA, Australia. [11]Northcott Neuroscience Laboratory, ANZAC Research Institute, Sydney, NSW 2139, Australia. [12]Faculty of Medicine and Health, University of Sydney, Sydney, NSW 2006, Australia. [13]Population Health and Immunity Division, The Walter and Eliza Hall Institute of Medical Research, 1 G Royal Parade, Parkville, VIC 3052, Australia. [14]Department of Medical Biology, The University of Melbourne, 1G Royal Parade, Parkville VIC3052, Australia. [15]Department of Mathematics and Statistics, Curtin University, Perth, WA, Australia. [16]Bruce Lefroy Centre, Murdoch Children's Research Institute, Parkville, VIC, Australia. [17]Department of Paediatrics, University of Melbourne, Royal Children's Hospital, Parkville, VIC, Australia. [18]Department of Neuropathology, National Hospital for Neurology and Neurosurgery, London, United Kingdom. [19]IRCCS Mondino Foundation, Pavia, Italy. [20]Centre de Référence des Maladies Neuromusculaires Nord-Est-Ile de France, Hôpital Pitié-Salpêtrière, Institut de Myologie, APHP, Paris, France. [21]CHU Grenoble Alpes, Grenoble Institut Neurosciences, INSERM, U1216, Université Grenoble Alpes, Grenoble, France. [22]Neurology Department, The Alfred Hospital, Melbourne, VIC, Australia. [23]The Laboratory for Genomic Medicine, University of Otago, Dunedin, New Zealand. [24]Molecular Medicine Laboratory, Concord Repatriation General Hospital, Sydney, NSW 2139, Australia. [25]Brain and Nerve Research Centre, Concord Repatriation General Hospital, Sydney, NSW 2139, Australia. [26]Department of Neurology, Royal Brisbane & Women's Hospital, Herston, QLD, Australia. [27]UQ Centre for Clinical Research, Herston, QLD, Australia. [28]Pathology Queensland, Royal Brisbane and Women's Hospital, Herston, QLD, Australia. [29]School of Biomedical Sciences, The University of Queensland, St. Lucia, QLD, Australia. [30]PathWest Neuropathology, Royal Perth Hospital, Perth, WA, Australia. [31]School of Medicine and Pharmacology, University of Western Australia, Crawley, WA, Australia. [32]Perron Institute for Neurological and Translational Science, Nedlands, WA, Australia. [33]Neurogenetics Unit, Diagnostic Genomics, PathWest, Nedlands, WA, Australia. [34]Dubowitz Neuromuscular Centre, UCL Great Ormond Street Institute of Child Health & MRC Centre for Neuromuscular Diseases, London, United Kingdom. [35]Wessex Neurological Centre, University Hospital Southampton, Southampton, United Kingdom. [36]William Harvey Research Institute, Faculty of Medicine and Dentistry, Queen Mary University of London, London, United Kingdom. [37]Department of Medical Biology, The University of Melbourne, Parkville, Victoria, Australia. [38]Department of Anatomical Pathology, Alfred Hospital, Melbourne, Victoria, Australia. [39]Neurogenetics Unit, Royal Perth Hospital, Perth, WA, Australia. [40]Department of Paediatrics Monash Children's Hospital, Victoria, Australia. [61]These authors contributed equally: Andrea Cortese, Sarah J. Beecroft. [62]These authors jointly supervised this work: Enrico Bugiardini, Gianina Ravenscroft. *A list of authors and their affiliations appears at the end of the paper. ✉e-mail: andrea.cortese@ucl.ac.uk; gina.ravenscroft@uwa.edu.au

## OPDM study group

Piraye Oflazer[41], Nazli A. Başak[41], Hülya Kayserili[41], Gözde Yeşil[42], Edoardo Malfatti[43,44], James B. Lilleker[45], Matthew Wicklund[46], Robert D. S. Pitceathly[1], Stefen Brady[47], Bernard Brais[48], David Pellerin[1,48], Stephan Zuchner[49], Matt C. Danzi[49], Marina Grandis[50,51], Giacomo P. Comi[52,53], Stefania P. Corti[52,54], Elena Abati[52,53], Antonio Toscano[55], Arianna Manini[52], Arianna Ghia[1,2], Cristina Tassorelli[2,19], Ilaria Quartesan[2], Roberto Simone[1], Alexander M. Rossor[1], Mary M. Reilly[1], Liam Carroll[35], Volker Straub[56], Bjarne Udd[57,58], Zhiyong Chen[59] & Gisèle Bonne[60]

[41]Koc University School of Medicine, Department of Neurology, Istanbul, Turkey. [42]Department of Medical Genetics, Istanbul Medical Faculty, Istanbul, Turkey. [43]Centre de Référence de Pathologie Neuromusculaire Nord-Est-Ile-de-France, Hôpital Henri Mondor, APHP, Créteil, France. [44]Université Paris Est, U955, INSERM, IMRB, F-94010 Créteil, France. [45]Manchester Centre for Clinical Neurosciences, Salford Royal Hospital, Northern Care Alliance NHS Foundation Trust, Manchester, UK. [46]UT Health San Antonio, San Antonio, TX, USA. [47]Oxford Muscle Service, John Radcliffe Hospital, Level 3, West Wing, Oxford OX3 9DU, UK. [48]Department of Neurology and Neurosurgery, Montreal Neurological Hospital and Institute, McGill University, Montreal, QC, Canada. [49]Dr. John T. Macdonald Foundation, Department of Human Genetics and John P. Hussman Institute for Human Genomics, University of Miami, Miller School of Medicine, Miami, FL, USA. [50]IRCCS Policlinico San Martino Hospital, Genoa, Italy. [51]Dipartimento Di Neuroscienze, Riabilitazione, Oftalmologia, Genetica e Scienze Materno-Infantili, Università Di Genova, Genoa, Italy. [52]Dino Ferrari Center, Department of Pathophysiology and Transplantation, University of Milan, 20122 Milan, Italy. [53]Neurology Unit, Fondazione IRCCS Ca' Granda Ospedale Maggiore Policlinico, Milan, Italy. [54]Neuromuscular and Rare Disease Unit, Fondazione IRCCS Ca' Granda Ospedale Maggiore Policlinico, Milan, Italy. [55]ERN-NMD Center for Neuromuscular Disorders of Messina, Department of Clinical and Experimental Medicine, University of Messina, Messina, Italy. [56]John Walton Muscular Dystrophy Research Centre, Newcastle University Translational and Clinical Research Institute and Newcastle Hospitals NHS Foundation Trust, Newcastle upon Tyne, UK. [57]Tampere Neuromuscular Center, Tampere, Finland. [58]Folkhalsan Research Center, Helsinki, Finland. [59]Department of Neurology, National Neuroscience Institute, Singapore 308433, Singapore. [60]Sorbonne Université, INSERM, Institut de Myologie, Centre de Recherche en Myologie, Paris, France.

