## [Peer Review File · Nature Communications]

A CCG expansion in ABCD3 causes oculopharyngodistal myopathy in individuals of European ancestryReviewer #1 (Remarks to the Author):

The authors reported novel CCG repeat expansions in ABCD3 associated with affected individuals from unrelated OPDM families of European descent, including two large Australian families. To analyze case samples, they used linkage analysis and short-read whole genome/exome sequencing to narrow down a candidate region and then used targeted ONT sequencing to identify CCG expansions from the region. Meanwhile, they used the 100,000 Genomics England Genome Project dataset (short-read sequencing) as a control. Although some of the reported findings are very interesting, there are many problems with the manuscript that prevent it from being published in its current form.

Comments:

In Figure 1A, the image of the family is difficult to see. Use high-resolution images to show numbers (such as I, II, III,..., 1, 2, 3,...) more clearly. FR1-II:2 is displayed in Figure 2D but is missing the family.

Figure 2B shows ABCD3 CCG repeat expansions with counts >60 are plotted in both myopathy cases and controls. However, I am skeptical of this analysis because these extensions exceed 180 bps and are difficult to observe using srWGS (short read whole genome sequencing). These long expansions need to be confirmed by long-read sequencing; in particular, the following analysis in L187-9 is done by srWGS:

"We next performed a more accurate profiling of the repeat locus using ExpansionHunter v3.2.2 which confirmed the presence of a large monoallelic CCG expansion of ABCD3 in two cases diagnosed with OPDM (estimates of 109 and 98 repeats) (Figure2B)."

The counts of these repeats are unreliable because UK1-II:1 and UK2-III:1 are estimated to be 98 and 109 repeats according to srWGS in Figure 2B, but ~380 and ~220 repeats according to the target nanopore long read sequence in Figure 3A, indicating a discrepancy between the srWGS and long-read sequences.

It is mentioned that DNA from UK2-II:1 was unavailable in L.193; however, Figure 2C shows bionano data from UK2-II:1. In L.199, they state that "in UK1-II:1, UK2-III:1, UK2-III:2 and FR1-II:1 fresh blood was available for extraction of ultralong fragments."; however, UK2-II:1 is missing in the list. Figure 2C shows the length difference between the control and UK2-II:1, which is expected to confirm the length difference observed by short-read sequencing in Figure 2B, but UK2-II:1 is not shown in Figure 2B. This analysis is completely confusing.

Figure 2D shows RP-PCR data of FR1-II:2; however, FR1-II:2 is missing in the pedigree in Figure 1A. FR1-II:2 cannot be found in the list of the following statement in L.197-9:

"We identified three additional probands of French nationality from unrelated families diagnosed with OPDM (FR1-II:1, FR2-II:2, FR3-II:1) carrying the CCG expansion in ABCD3 (Figure 2D)."

In L.217-224, the authors should describe the discrepancy in the length of CCG repeat expansions observed by srWGS and Nanopore long-read sequencing. As described before, you should state that UK1-II:1 and UK2-III:1 are estimated to have 98 and 109 repeats according to srWGS but they are ~380 and ~220 repeats according to targeted Nanopore long-read sequencing in Figure 3A. Discuss the reliability of Figure 2B using srWGS data.

Show the precise measure of the x-axis (CCG trinucleotide repeats) in Figure 3A. Which is the number of bases or the number of CCG repeat units?

Figure 3A needs to be revised substantially. For example, look at the first row named AUS1-IV:3. We can find 26 occurrences of yellow "G"; however, the value in the x-axis is ~140. Does this number represent 26 units or 78 bases? Neither case indicates ~140. The other rows have similar problems.

In L.227, it is claimed that FR3-I:2 had the largest expansion of 624 repeats, but in Figure 3A, FR3-I:2 has >680 repeats. Why is such a large difference observed?

In L.235, n=6 is very small. Clearly explain what statistical methods are being used. Did you use a one-tailed test? If you specify a p-value, explain the statistical method used to derive the p-value.

In the paragraph starting from L.251 and in Figure 3C, you only showed eight "affected" individuals. In Figure 3C, you should display a comparison of haplotypes between control samples and affected ones in order to clearly state that the haplotype shared among the eight affected samples are nearly absent (0.2% in L.257) in the control samples.

In the paragraph starting with L.298, did you observe nascent RNAs and see that RNA foci within the nucleus were enriched with those nascent RNAs? This is because the CCG repeats are found in 5'UTR. Have you observed transcriptional abortion at the CCG repeats? Is there a significant relationship between CpG methylation and ABCD3 expression?

Reviewer #2 (Remarks to the Author):

Cortese, Beecroft, and their colleagues described families with CCG repeat expansions in the ABCD3 gene, resulting in oculopharyngodistal myopathy. They identified a total of 35 patients from 8 families located in Australia, the UK, and France. The genetic analysis was generally well-executed and provided conclusive results, considering the number of patients and families involved.

In the title, they referred to the condition as "cranial and distal limb myopathy," which is a new term. However, the usage of "cranial" may be misleading since the disease primarily affects muscles and is not related to cranial neuropathy. It might be more appropriate to reconsider this terminology.

On line 96, they mentioned "GCC/CCG repeats," but "GCC" and "CCG" are essentially the same sequence. It appears that "CGG/CCG repeats" would be a more accurate description.

In the supplementary materials, there are instances of "X cases" remaining, and they referred to "Gleeson" in Figure S4, which should be corrected to "Gleason." Additionally, Figures S2 and S4 lack legends. These issues should be addressed to improve the manuscript's quality.

In the linkage analysis, the authors presented LOD scores, but these scores may not make sense without the conditions and parameters used in the parametric analysis. For instance, it's important to know how the affected status of IV-2 in the AUS1 family was considered and whether the authors adjusted for penetrance.

They also detected a common disease haplotype among the families, likely due to a founder effect considering geographic information. However, it's worth considering whether CCG repeats within this haplotype have a tendency to become longer. It would be valuable to know the length of the repeats in cases with the haplotype found in the UK Biobank samples.

In lines 244 through 249, the authors argue that the number of affected children may be lower when the disease is transmitted from the mother compared to transmission from the father. Firstly, it's important to consider whether the age of the children plays a role in this difference. If the children from maternal transmission tend to be younger, it might naturally result in a lower percentage of them developing the disease. Additionally, while the term "penetrance" is used, it's typically associated with describing the percentage of carriers who develop the disease. It would be helpful for the authors to clarify their use of the term in this context.

The presentation of detailed clinical information is valuable. However, it would be beneficial to know if muscle CT/MRI scans were performed or considered as part of the clinical evaluation. These scans can provide essential insights into the extent of muscle involvement and may further enhance our understanding of the disease's progression.

In Figure 6B, I can hardly see red signals in the PDF. Are these three images derived from patients' fibroblasts? Kindly provide clarification on this matter.

Reviewer #3 (Remarks to the Author):

Cortese et al. reported the identification of a CCG repeat expansion in ABCD3 gene causing OPDM, which is a form of distal muscular dystrophy whose genetic cause remained unknown in the non-Asian ancestry. The proposed manuscript is important not only because it facilitates the future diagnosis of OPDM families of European ancestry, but also because it expands our understanding of the shared genetic cause of this disease across different populations. Manuscript is interesting, well written, and important to understand pathogenic mechanisms of OPDM. However, there are several issues that should be addressed appropriately:

Major concerns:

- 1) The author performed RP-PCR on ABCD3-OPDM patients and confirmed the segregation in the family UK2. RP-PCR was also performed to confirm the CCG repeat expansion in other OPDM patients. Is it possible to perform PCR fragment analysis on all the ABCD3-OPDM patients in this study? So, we can have a better understanding of the pathogenic range of this gene in OPDM.
- 2) Two affected females harbouring large expansions of the expanded allele was hypermethylated. The methylation was detected in blood DNA. How about the repeat size and methylation level in muscle samples?
- 3) In the muscle pathology, the authors claimed that rare p62-positive intranuclear inclusions were found. How about the p62 immunofluorescence staining on muscle biopsies? It should be more sensitive than IHC. How about Ub positive inclusions in muscle?
- 4) In Fig 5, the haematoxylin and eosin staining in panel A is not a typical RV. It is better to change it. Modified Gomori trichrome staining in panel B also has the same issue. They claimed the p62-positive (red) intra-nuclear inclusion in cultured primary skin 533 fibroblast from an OPDM individual in panel M. However, the panel M was not shown.
- 5) In table 1, they presented the clinical features of 24 affected individuals from eight families. Is it possible to add a table to compare the clinical characteristics among ABCD3-OPDM and other forms of OPDM, such as OPDM1 to OPDM4?
- 6) They performed analysis of skeletal muscle RNA-seq on three ABCD3-OPDM individuals. It is better performed qPCR to confirm the higher expression level of ABCD3 in these patients. At the same time, the expression level of ABCD3 antisense strand should be presented by qPCR.
- 7) The author performed RNA FISH on fibroblasts. It is better to show RNA foci in muscle biopsies. Thus, RNA FISH on ABCD3-OPDM muscle samples should be performed.

Minor concerns:

- 1) For the title of this manuscript, the authors used "cranial and distal limb myopathy" instead of "OPDM". In the manuscript, they claim the patients were all OPDM. Is it possible to make the title more specific?
- 2) In Fig 2D, the Y-axis should be showed in the RP-PCR data.
- 3) In the discussion part, it is suggested to add some discussion on the CGG repeat expansion related diseases and the pathogenic mechanisms of these diseases, such as FMR1 related FXTAS, NOTCH2NLC related NIID/ OPDM3.

POINT BY POINT RESPONSE TO THE REVIEWERS' COMMENTS

REVIEWER COMMENTS

Reviewer #1 (Remarks to the Author):

The authors reported novel CCG repeat expansions in ABCD3 associated with affected individuals from unrelated OPDM families of European descent, including two large Australian families. To analyze case samples, they used linkage analysis and short-read whole genome/exome sequencing to narrow down a candidate region and then used targeted ONT sequencing to identify CCG expansions from the region. Meanwhile, they used the 100,000 Genomics England Genome Project dataset (short-read sequencing) as a control. Although some of the reported findings are very interesting, there are many problems with the manuscript that prevent it from being published in its current form.

Comments:

In Figure 1A, the image of the family is difficult to see. Use high-resolution images to show numbers (such as I, II, III,..., 1, 2, 3,...) more clearly. FR1-II:2 is displayed in Figure 2D but is missing the family. *(A) We have made the pedigrees and numbers more clear. Figure 1 now only shows the pedigrees of OPDM individuals enrolled in this study, while the linkage analysis plot has been moved to supplementary data (Supplementary Figure S1)*
(B) Figure 2D is for individual FR2-II:1. Apologies for this error, this has been corrected.

Figure 2B shows ABCD3 CCG repeat expansions with counts >60 are plotted in both myopathy cases and controls. However, I am skeptical of this analysis because these extensions exceed 180 bps and are difficult to observe using srWGS (short read whole genome sequencing). These long expansions need to be confirmed by long-read sequencing; in particular, the following analysis in L187-9 is done by srWGS:

“We next performed a more accurate profiling of the repeat locus using ExpansionHunter v3.2.2 which confirmed the presence of a large monoallelic CCG expansion of ABCD3 in two cases diagnosed with OPDM (estimates of 109 and 98 repeats) (Figure2B).”

The counts of these repeats are unreliable because UK1-II:1 and UK2-III:1 are estimated to be 98 and 109 repeats according to srWGS in Figure 2B, but ~380 and ~220 repeats according to the target nanopore long read sequence in Figure 3A, indicating a discrepancy between the srWGS and long-read sequences.

We agree with the reviewer about the limitation of srWGS in accurately sizing repeats larger than the read length itself. We have added the following sentence “Notably, EH analysis performed on srWGS data tended to underestimate the actual size of repeat expansions in subjects who also underwent targeted LRS (98 vs 381 repeats in UK1-II:1 and 109 vs 231 in UK2-III:1). This observation confirms previous understanding that, while EH may suggest the presence of large expansions based on srWGS, it is not reliable for their accurate sizing, particularly when expansions are larger than the sequencing read length (~150 nucleotides).”

It is mentioned that DNA from UK2-II:1 was unavailable in L.193; however, Figure 2C shows bionano data from UK2-II:1. In L.199, they state that “in UK1-II:1, UK2-III:1, UK2-III:2 and FR1-II:1 fresh blood was available for extraction of ultralong fragments.”; however, UK2-II:1 is missing in the list. Figure 2C shows the length difference between the control and UK2-II:1, which is expected to confirm the length difference observed by short-read sequencing in Figure 2B, but UK2-II:1 is not shown in Figure 2B. This analysis is completely confusing.

We are truly sorry about the confusion.

Figure 2C represents data from UK2-III-1 and not UK2-II-1, while Figure 2B is correct. UK2-II-1 is deceased, and no blood or DNA was available for testing.

Figure 2D shows RP-PCR data of FR1-II:2; however, FR1-II:2 is missing in the pedigree in Figure 1A. FR1-II:2 cannot be found in the list of the following statement in L.197-9:

“We identified three additional probands of French nationality from unrelated families diagnosed with OPDM (FR1-II:1, FR2-II:2, FR3-II:1) carrying the CCG expansion in ABCD3 (Figure 2D).”

We would like to apologise again for the typos and we thanks the reviewer for the careful revision. FR1-II-2 should be FR1-II-1. This was corrected in figure 2D.

In L.217-224, the authors should describe the discrepancy in the length of CCG repeat expansions observed by srWGS and Nanopore long-read sequencing. As described before, you should state that UK1-II:1 and UK2-III:1 are estimated to have 98 and 109 repeats according to srWGS but they are ~380 and ~220 repeats according to targeted Nanopore long-read sequencing in Figure 3A. Discuss the reliability of Figure 2B using srWGS data.

As discussed above the following sentence was added : “Notably, EH analysis performed on srWGS data tended to underestimate the actual size of repeat expansions in subjects who also underwent targeted LRS (98 vs 381 repeats in UK1-II:1 and 109 vs 231 in UK2-III:1). This observation confirms previous understanding that, while EH may suggest the presence of large expansions based on srWGS, it is not reliable for their accurate sizing, particularly when expansions are larger than the sequencing read length (~150 nucleotides).”

Show the precise measure of the x-axis (CCG trinucleotide repeats) in Figure 3A. Which is the number of bases or the number of CCG repeat units?

The number of CCG repeats are now shown along the x-axis of the figure.

Figure 3A needs to be revised substantially. For example, look at the first row named AUS1-IV:3. We can find 26 occurrences of yellow “G”; however, the value in the x-axis is ~140. Does this number represent 26 units or 78 bases? Neither case indicates ~140. The other rows have similar problems.

We have corrected the issue with compression of the image, apologies for the confusion this compression of the figure caused in trying to make sense of the data.

In L.227, it is claimed that FR3-I:2 had the largest expansion of 624 repeats, but in Figure 3A, FR3-I:2 has >680 repeats. Why is such a large difference observed?

This is a typographical error, the largest expansion is 694 repeats and we have corrected this in the manuscript.

In L.235, n=6 is very small. Clearly explain what statistical methods are being used. Did you use a one-tailed test? If you specify a p-value, explain the statistical method used to derive the p-value.

There were 8 samples sequenced by long-read from affected males, however for 2 of these we did not know the age of onset, so only 6 males were included in the linear regression for repeat size vs. age of onset. For the comparison of repeat sizes in affected males and females, a two-tailed t-test was performed, these methods are in the supplementary material.

In the paragraph starting from L.251 and in Figure 3C, you only showed eight “affected” individuals. In Figure 3C, you should display a comparison of haplotypes between control samples and affected ones in order to clearly state that the haplotype shared among the eight affected samples are nearly absent (0.2% in L.257) in the control samples.

A representative haplotype from a control sample was added on Figure 3C.

In the paragraph starting with L.298, did you observe nascent RNAs and see that RNA foci within the nucleus were enriched with those nascent RNAs? This is because the CCG repeats are found in 5'UTR.

This is an interesting point and we hope we understood the comment correctly.

We did not check specifically for the presence of ABCD3 pre-mRNA in the nuclear foci observed in patients' muscle or fibroblasts.

However, as shown in the figure attached below, CCG expansions are included in the mature ABCD3 transcript, according to RNAseq data. Therefore, we think that FISH for mature ABCD3 transcript can be used to test for the presence of repeat containing RNA foci.

CCG STR locus

Detection of repeat containing foci through traditional FISH targeting the repeat itself (and flanking sequence) works well in case of very large expansion (eg myotonic dystrophy type 2) but is less effective in the case of smaller expansions or low abundant transcript (Glineburg, M.R. et al. Acta Neuropathol Commun 9, 73 (2021). The Hybridisation Chain Reaction (HCR) RNA Fluorescence in situ hybridisation (FISH) method we have used here leverages 20 probes spanning the entire ABCD3 gene (from the repeat containing 5' UTR to the 3' end) and enables a better visualization of the RNA molecules.

We have not performed nascent RNA sequencing, which would only be possible on cell lines (e.g. fibroblasts) but would hardly provide any additional spatial information on the enrichment of nascent RNA in nuclear ABCD3 foci in affected muscle tissue.

Have you observed transcriptional abortion at the CCG repeats?

Although the accurate assessment of transcriptional abortion may require additional methods (eg classic Maxam and Gilbert sequencing or nascent RNA seq, see Aaron R Haeusler et al, C9orf72 nucleotide repeat structures initiate molecular cascades of disease, Nature. 2014;507(7491):195-200. doi: 10.1038/nature13124), RNAseq did not show a different coverage of 5' vs 3' end of the gene. Also, the gene is overall upregulated so that the CCG expansion in the promoter/5UTR region does not appear to hinder transcription of the gene. Therefore, we assume that truncated mRNA ABCD3 transcript are not significantly enriched in ABCD3 OPDM, but we agree that we cannot rule this out either.

Is there a significant relationship between CpG methylation and ABCD3 expression?

We agree it is possible that an increased CpG methylation may lead, similarly to CGG expansion in FMR1, to gene silencing. However, unfortunately muscle tissue or cell lines from the 2 subjects showing hypermethylation of ABCD3 5'UTR/promoter (FR1-II-1 and FR3-I-2) was not available to test this hypothesis. We have added this information in results "Unfortunately, muscle tissue or cell lines were not available from the two subjects showing hypermethylated expanded alleles (FR1-II-1 and FR3-I-2) to test this hypothesis. "

Reviewer #2 (Remarks to the Author):

Cortese, Beecroft, and their colleagues described families with CCG repeat expansions in the ABCD3 gene, resulting in oculopharyngodistal myopathy. They identified a total of 35 patients from 8 families located in Australia, the UK, and France. The genetic analysis was generally well-executed and provided conclusive results, considering the number of patients and families involved.

Thank you for this positive critique of our study.

In the title, they referred to the condition as "**cranial and distal limb myopathy,**" which is a new term. However, the usage of "cranial" may be misleading since the disease primarily affects muscles and is not related to cranial neuropathy. It might be more appropriate to reconsider this terminology. *We have modified the title to "A CCG expansion in ABCD3 causes oculopharyngodistal myopathy in individuals of European ancestry"*

On line 96, they mentioned "GCC/CCG repeats," but "GCC" and "CCG" are essentially the same sequence. It appears that "CGG/CCG repeats" would be a more accurate description.

GCC • CCG was replaced by CGG • CCG throughout the text.

In the supplementary materials, there are instances of "X cases" remaining, and they referred to "Gleeson" in Figure S4, which should be corrected to "Gleason." Additionally, Figures S2 and S4 lack legends. These issues should be addressed to improve the manuscript's quality.

The Gleason typographical error has been corrected.

In the linkage analysis, the authors presented LOD scores, but these scores may not make sense without the conditions and parameters used in the parametric analysis. For instance, it's important to know how the affected status of IV-2 in the AUS1 family was considered and whether the authors adjusted for penetrance.

For the linkage studies we did not adjust for penetrance since the families we did linkage on initially appeared to have a fully penetrant AD inheritance of OPDM. All individuals included in the PED file were either marked as affected or unaffected, since AUS1-IV:2 has 2 affected children, we marked him as affected for the purposes of the linkage. He died in his 20s from a motorcycle accident – this is before the age of onset typically seen in his relatives.

We used the following parameters, which are standard for AD disease:

Disease allele frequency: 0.01

Penetrance (Probability of being affected with):

0 alleles: 0.001

1 allele: 1.0

2 alleles: 1.0

They also detected a common disease haplotype among the families, likely due to a founder effect considering geographic information. However, it's worth considering whether CCG repeats within this haplotype have a tendency to become longer. It would be valuable to know the length of the repeats in cases with the haplotype found in the UK Biobank samples.

While we did not have access yet to UK Biobank samples, we were able to analyse the entire Rare Diseases cohort of Genomics England (allele numbers= 69,358) and showed that the expansion is larger in alleles corresponding to the disease-associated haplotype. We added the following sentence "The shared haplotype lies primarily within a low-recombination region (HapMap data) and has an allele frequency of 0.13% in the 100,000 Genome Project (Rare Disease Cohort). Within the Rare Disease cohort (number of alleles analysed = 69,358), we analysed the repeat count estimation by ExpansionHunter in individuals carrying the disease-associated haplotype and in non-carriers. Notably, while expanded alleles are observed in both groups, the repeat count was higher in subjects carrying the disease-associated haplotype (median=30 repeats, IQR=27-34) vs other haplotypes (median=7 repeats, IQR=7-7), with a p-value of 1.5e-264 (p<0.0001) for the Mann-Whitney test, suggesting that it may represent a more permissive haplotype for the occurrence of large expansions (Figure 3D).

In lines 244 through 249, the authors argue that the number of affected children may be lower when the disease is transmitted from the mother compared to transmission from the father. Firstly, it's important to consider whether the age of the children plays a role in this difference. If the children from maternal transmission tend to be younger, it might naturally result in a lower percentage of them developing the disease. Additionally, while the term "penetrance" is used, it's typically associated with describing the percentage of carriers who develop the disease. It would be helpful for the authors to clarify their use of the term in this context.

The unaffected children of affected women are of similar ages to that of their affected cousins (born to affected uncles). For example, the 12 affected children born to individuals AUS2-IV:2, IV:3, IV:4 and IV:12 are between 29 and 61 years of age, with an average age of 50; none of these individuals have any clinical features of the disease.

Incomplete penetrance is documented in other genetic forms of OPDM (see review Kumutponpanich, T., and T. Liewluck. 2022. 'Oculopharyngodistal myopathy: The recent discovery of an old disease', Muscle Nerve, 66: 650-52) and refers to the observation that some individuals who harbour pathogenic expansions do not develop OPDM. For clarity we have modified reduced penetrance to incomplete penetrance in the manuscript.

The presentation of detailed clinical information is valuable. However, it would be beneficial to know if muscle CT/MRI scans were performed or considered as part of the clinical evaluation. These scans can provide essential insights into the extent of muscle involvement and may further enhance our understanding of the disease's progression.

We agree with the reviewer and MRI scan from patient UK2-III-2 were added to Figure 4 and finding were commented in the text "Lower limbs muscle MRI was available for one UK proband (UK2-III-2) (Figure 4 L and M). Pattern of involvement was in keeping with the clinical manifestations showing a marked predominant involvement of the distal muscles (mostly soleus and gastrocnemius) compared to the thigh muscles that were mostly preserved. "

Unfortunately, muscle MRI and CT scans are not available for any of the other affected individuals.

In Figure 6B, I can hardly see red signals in the PDF. Are these three images derived from patients' fibroblasts? Kindly provide clarification on this matter.

We have improved the quality and labelling of figure 6B and we have also added representative images and quantification plot from ABCD3 FISH from muscle biopsy, further supporting the presence of increased ABCD3 nuclear signal / foci in OPDM samples.

Reviewer #3 (Remarks to the Author):

Cortese et al. reported the identification of a CCG repeat expansion in ABCD3 gene causing OPDM, which is a form of distal muscular dystrophy whose genetic cause remained unknown in the non-Asian ancestry. The proposed manuscript is important not only because it facilitates the future diagnosis of OPDM families of European ancestry, but also because it expands our understanding of the shared genetic cause of this disease across different populations. Manuscript is interesting, well written, and important to understand pathogenic mechanisms of OPDM. However, there are several issues that should be addressed appropriately:

Major concerns:

1) The author performed RP-PCR on ABCD3-OPDM patients and confirmed the segregation in the family UK2. RP-PCR was also performed to confirm the CCG repeat expansion in other OPDM patients. Is it possible to perform PCR fragment analysis on all the ABCD3-OPDM patients in this study? So, we can have a better understanding of the pathogenic range of this gene in OPDM.

We have a really good idea of the pathogenic range (repeat size) from the long-read sequencing we have done on 19 OPDM patients. This is as many OPDM patients as possible since DNA from other affected individuals was of insufficient quantity or quality for long-read sequencing. Nor RP-PCR or sizing/flanking PCR are able to accurately size the repeats, due to the expansion size being larger than the detection limit of the assay.

2) Two affected females harbouring large expansions of the expanded allele was hypermethylated. The methylation was detected in blood DNA. How about the repeat size and methylation level in muscle samples?

Unfortunately, muscle tissue from the two subjects showing hypermethylation of ABCD3 promoter in blood (FR1-II-1 and FR3-I-2). We have added this information in results "Unfortunately, muscle tissue or cell lines were not available from the two subjects showing hypermethylated expanded alleles (FR1-II-1 and FR3-I-2)."

3) In the muscle pathology, the authors claimed that rare p62-positive intranuclear inclusions were found. How about the p62 immunofluorescence staining on muscle biopsies? It should be more sensitive than IHC. How about Ub positive inclusions in muscle?

Immunofluorescence was performed for p62, there were no p62-positive intranuclear inclusions seen in any of the three patient muscle biopsies examined; we have added an image of this to Figure 5 (Figure 5I).

4) In Fig 5, the haematoxylin and eosin staining in panel A is not a typical RV. It is better to change it. Modified Gomori trichrome staining in panel B also has the same issue. They claimed the p62-positive (red) intra-nuclear inclusion in cultured primary skin 533 fibroblast from an OPDM individual in panel M. However, the panel M was not shown.

We have updated the figure to include clearer images of the pathology. Our apologies, the p62 inclusion in skin was image L in the panel, it is now included as image 5O.

5) In table 1, they presented the clinical features of 24 affected individuals from eight families. Is it possible to add a table to compare the clinical characteristics among ABCD3-OPDM and other forms of OPDM, such as OPDM1 to OPDM4?

A Supplementary table comparing the frequency of key clinical features of OPDM1-5 has been added.

6) They performed analysis of skeletal muscle RNA-seq on three ABCD3-OPDM individuals. It is better performed qPCR to confirm the higher expression level of ABCD3 in these patients. At the same time, the expression level of ABCD3 antisense strand should be presented by qPCR.

As detailed in our supplementary methods, we generated strand-specific (reverse) paired-end RNAseq data, which allows for one to differentiate between sense and antisense reads. In brief, we colour the reads in IGV by "first-of-pair-strand", which allows you to separate sense and antisense reads if you have stranded RNAseq (which we do). The red is forward (+) and the blue is reverse (-). ABCD3 is on the positive strand (the arrows in the gene point to the right), but the reads are blue (minus strand), this is because it is a REVERSE stranded library (which appears to be more common than FORWARD stranded). From this analysis, there were only sense reads containing the expanded repeat sequence on RNAseq from the three affected.

Below is a snapshot taken in IGV showing the reads mapping to the 5'UTR of ABCD3 in OPDM muscle RNA-seq data, the vast majority of reads are for the sense strand and only sense strand reads contain repeat expansions.

Moreover, no evidence of ABCD3 antisense transcript was found in public available databases including Cap Analysis of Gene Expression (CAGE) data from the FANTOM5 project (Supplementary Figure S6).

For improved clarity, CAGE is a technique which allows to map a short sequence at the 5' end of a transcript, tagging the Transcription Start Sites (TSSs) and providing a quantification of transcript

abundances in a sample. As shown in Supplementary Figure S6, no antisense CAGE signal is detected for ABCD3 in any tissue.

We added the following Figure and legend in supplementary data:

Figure S6. FANTOM5 CAGE data for ABCD3. In the top pane, the ABCD3 MANE transcript is highlighted in pink. In the middle pane, CAGE signal for the sense (green) and antisense (purple) direction. In the bottom pane, main CAGE peaks (detected only in the sense direction). Data are taken from the Human hg38 Promoterome in FANTOM5 ZENBU website, [<https://fantom.gsc.riken.jp/zenbu/>].

We also added in results the following sentence “Moreover, no evidence of ABCD3 antisense transcript was found in our RNAseq data, nor from public available databases including Cap Analysis of Gene Expression (CAGE) data from the FANTOM5 project (Supplementary Figure S6)”.

We also conducted qPCR for ABCD3 expression in available muscle biopsies and included these data as Supplementary Figure S5.

7) The author performed RNA FISH on fibroblasts. It is better to show RNA foci in muscle biopsies. Thus, RNA FISH on ABCD3-OPDM muscle samples should be performed.

We were able to obtain fresh frozen slides from 1 ABCD3-OPDM patient and we performed RNA FISH. We observed an increased nuclear signal for ABCD3 transcript in ABCD3 OPDM vs control. We have included representative images and a quantification plot as **Figure 6B-C**

We have modified the text accordingly “..we next performed HCR™ RNA-FISH for the ABCD3 sense transcript on fibroblasts and frozen skin and muscle sections from affected OPDM individuals. We identified increased cytoplasmic and intranuclear signal of ABCD3 transcript in patient-derived fibroblasts (UK2-III:2) and muscle tissue (AUS2-IV:4) compared to controls (**Figure 6B**). This is

congruent with increased ABCD3 transcript expression detected in skeletal muscle RNA-seq. In the muscle biopsy the ABCD3 signal appeared more clustered in nuclei to form foci like structures. ABCD3-positive foci were also identified in the skin biopsy of one OPDM individual but were exceedingly rare (one out of >100 nuclei, **Supplementary Figure S4**).” (results)

Supplementary methods “We performed RNA in situ hybridisation for ABCD3 transcript on human derived skin fibroblasts from one patient (UK2-III:2) and three age-matched controls and on frozen muscle sections from one individual (AUS2-IV:4) and one control” and “For frozen muscle sections, samples were processed according to Molecular Instrument’s protocol. Briefly, slides were fixed in 4% paraformaldehyde for 15 minutes at 4°C and subsequently dehydrated with a graded series of ethanol. After two washes in PBS, sections were pre-heated in probe hybridization buffer (Molecular Instruments) for 10 minutes at 37°C and then incubated with 1.6 pmol of ABCD3 probe set in hybridisation buffer at 37°C overnight. The following day slides were washed in four times in probe wash buffer and gradually increasing concentrations of 5x SSC-T at 37°C and once in 5x SSC-t at room temperature. The amplification stage was performed as described above for the skin fibroblasts. After three washes in 5x SSC-T at room temperature, slides were treated with 0.1% Sudan Black in ethanol 70% and washed thrice in ethanol 30%. Finally, slides were counterstained with DAPI for 15 minutes and then mounted with Dako mounting medium.”

Minor concerns:

1) For the title of this manuscript, the authors used “cranial and distal limb myopathy” instead of “OPDM”. In the manuscript, they claim the patients were all OPDM. Is it possible to make the title more specific?

We have modified the title of the study to “A CCG expansion in ABCD3 causes oculopharyngodistal myopathy in individuals of European ancestry”

2) In Fig 2D, the Y-axis should be showed in the RP-PCR data.

The Y axis and label (fluorescence intensity) was added to Fig 2D.

3) In the discussion part, it is suggested to add some discussion on the CGG repeat expansion related diseases and the pathogenic mechanisms of these diseases, such as FMR1 related FXTAS, NOTCH2NLC related NIID/ OPDM3.

We have added additional text to the discussion regarding the pathomechanisms of these similar disorders.

Reviewer #1 (Remarks to the Author):

The author answered most of my questions satisfactorily and corrected major typos in the previous version.

Due to image compression issues, it is still difficult to see the details in Figures 1 and 3A. The authors should provide higher resolution images of these two figures.

Reviewer #2 (Remarks to the Author):

The authors generally responded well to the reviewers' comments.

In the discussion, they wrote "80-200 CGG repeats in the 5' UTR of FMR1", which I think usually considered to be 55-200 repeat.

Reviewer #3 (Remarks to the Author):

Thank you for your detailed reply. All my concerns have been addressed.

RESPONSE TO REVIEWERS' COMMENTS

We thank all reviewer for their insightful comments and feedback which have significantly improved the manuscript

Reviewer #1 (Remarks to the Author):

The author answered most of my questions satisfactorily and corrected major typos in the previous version.

Due to image compression issues, it is still difficult to see the details in Figures 1 and 3A. The authors should provide higher resolution images of these two figures.

Higher quality images were provided

Reviewer #2 (Remarks to the Author):

The authors generally responded well to the reviewers' comments.

In the discussion, they wrote "80-200 CGG repeats in the 5' UTR of FMR1", which I think usually considered to be 55-200 repeat.

The sentence was amended to "55-200 CGG repeats in the 5' UTR of FMR1"

Reviewer #3 (Remarks to the Author):

Thank you for your detailed reply. All my concerns have been addressed.